

# Lateral heat fluxes amplify the aggregation error of soil temperature in non-sorted circles

Melanie A. Thurner[1,2], Xavier Rodriguez-Lloveras[1], and Christian Beer[1]

[1]Institute of Soil Science, University of Hamburg, Hamburg, Germany
[2]KIT-Campus Alpin, Garmisch-Partenkirchen, Germany

**Correspondence:** Melanie A. Thurner (melanie.a.thurner@web.de)

**Abstract.** Soil properties vary within centimeters, which is not captured by state-of-the-art land-surface models due to their kilometer-scale grid. This mismatch can lead to systematic errors when simulating the exchange of energy, water, and greenhouse gases between the land and atmosphere—collectively referred to as "aggregation error."

To quantify the potential aggregation error of soil temperature, we developed the two-dimensional pedon-scale geophysical
soil model DynSoM-2D, which has a spatial resolution of 10cm. We applied DynSoM-2D at a permafrost-affected, non-sorted circle site using three different setups: (i) a homogeneous soil profile representing a typical land surface model, compiled by averaging the heterogeneous soil inputs; (ii) the actual heterogeneous soil profile of a typical non-sorted circle; and (iii) the heterogeneous soil profile including lateral heat fluxes.

Our results show that DynSoM simulates warmer soil temperatures when heterogeneous soil properties are considered, with this
warming becoming even more pronounced and consistent across the domain when lateral heat fluxes are included. By aggregating grid cells, we traced the aggregation error back to the spatial distribution of organic matter, which nonlinearly alters soil thermal and hydrological properties, leading to the observed differences between simulations. In our case, the heterogeneity-induced warming led to a deepening of the active layer and an extension of the snow-free period, both of which can strongly alter ecosystem dynamics, while having only a minor effect on soil-atmosphere heat exchange on an annual basis.

## 1 Introduction

Permafrost-affected ecosystems are unique due to their extreme climatic conditions, which result in a short vegetation season and characteristic soil patterns (Siewert et al., 2021; Schädel et al., 2024). The annual freeze-thaw cycle drives distinct three-dimensional soil movements, leading to pronounced surface and subsurface heterogeneity (Ping et al., 2015; Beer, 2016). These movements, collectively known as *cryoturbation*, give rise to typical patterned ground features such as non-sorted cir-
cles. These circles, which contain a mix of soil materials, span several meters in diameter, while their internal texture variability occurs on the centimeter scale (Ping et al., 2015; Siewert et al., 2021).

Soil texture and organic matter content play a crucial role in determining thermal and hydrological properties, thereby influencing processes such as water flow and the formation of segregated ice (Bockheim, 2007; Nicolsky et al., 2008). In turn, these processes alter particle distributions through frost heave and sediment transport, further modifying soil texture. This



bidirectional relationship between soil texture and dynamics is a hallmark of cryoturbation. It is particularly important when considering soil organic matter (OM), which is redistributed by cryoturbation from the topsoil into deeper layers (Ping et al., 2008, 2015). As a result, permafrost soils—especially turbel subsoils—often exhibit peaks in OM content at depth, a pattern that contrasts with the typical vertical decline seen in other soil types (Batjes, 1996; Jobbágy and Jackson, 2000; Harden et al., 2012; Gentsch et al., 2015; Beer et al., 2022). Since OM has distinct thermal and hydrological characteristics, its redistribution
feeds back into the same processes that drive cryoturbation (Nicolsky et al., 2008; He et al., 2021; Zhu et al., 2019), forming a tightly coupled system of biotic and abiotic interactions. This system is a defining feature of permafrost soils and is deeply rooted in, and perpetuates, small-scale heterogeneity (Nicolsky et al., 2008; Ping et al., 2015).

Beyond internal soil dynamics, heterogeneity also influences the exchange of energy, water, and greenhouse gases between the land surface and atmosphere (Lawrence et al., 2015; Bonan, 2019; Schädel et al., 2024). Variations in surface and subsurface
properties produce spatial patterns in heat fluxes that affect surface temperatures (Aalto et al., 2018). For instance, the texture of the soil and the topography of the land determine the seasonal snow cover and the height of vegetation, both of which vary with the season (Ekici et al., 2014; Aalto et al., 2018). Furthermore, the properties of the soil determine hydrological processes that affect the latent heat flux. Snow insulates the soil, with the effectiveness of this insulation increasing with snow height. Snow also exhibits high radiative reflectance, which is why the presence and amount of snow have a significant impact on the surface
energy balance (Ekici et al., 2014). Vegetation exerts a significant influence on exchange rates through various physiological processes, including water uptake and transpiration, which regulate the availability of water in the environment. Additionally, vegetation's unique capacity to absorb and reflect energy contributes to the regulation of energy balance within the ecosystem. The assimilation and release of carbon (C) is another crucial aspect of vegetation's impact on the environment, playing a pivotal role in the global C cycle (Miralles et al., 2025). As outlined in the work of Bonan (2019), the intricate relationship between
vegetation and C dynamics is a crucial factor in understanding and mitigating environmental impacts. The latter encompasses not only the direct C emission resulting from respiration, but also the indirect influence on the C dynamics within the soil. This indirect influence is characterised by the conversion of C into soil OM and the subsequent impact on the microbial community within the soil. The function of this community is to facilitate the decomposition of OM, a process that ultimately leads to the production of greenhouse gases, namely carbon dioxide ($CO_2$) and methane ($CH_4$) (Bonan, 2019).

Soil heterogeneity, therefore, has implications not only for site-specific soil processes but also for land–atmosphere interactions on regional and global scales (Beer, 2008; Ping et al., 2015; Nicolsky et al., 2008; Nitzbon et al., 2021). This is particularly relevant for modeling current and future ecosystem dynamics (Hagemann et al., 2016; Aas et al., 2019; Cai et al., 2020; Martin et al., 2021; Schädel et al., 2024). However, Arctic soil heterogeneity is generally not represented in state-of-the-art land surface models (LSMs), as the features occur at sub-grid scales (Beer, 2016; Burke et al., 2020; Schädel et al., 2024). Soil tex-
ture is typically aggregated across kilometer-scale grid cells, dramatically reducing heterogeneity (Rastetter et al., 1992). This can lead to distortions in model outputs, since small-scale processes may interact in non-linear ways that cannot be resolved explicitly. This issue is described as *aggregation error* (Rastetter et al., 1992). In addition to aggregation, there is uncertainty in the parameterization of small-scale permafrost processes, which further limits the pace of improvement in LSMs for Arctic





systems (Burke et al., 2020; Schädel et al., 2024).


To tackle this uncertainty, an increasing number of studies explores on the effect of permafrost and permafrost-affected soil representation in models. They range from site-level to global scales, e.g. Nicolsky et al. (2008); Westermann et al. (2016); Aas et al. (2019); Nitzbon et al. (2021); Smith et al. (2022); de Vrese et al. (2022), and cover various questions and approaches from general plausible representations of permafrost and their implications (de Vrese et al., 2022) over commonly used assumptions

and simplifications of permafrost-specific processes (Cai et al., 2020; Gao and Coon, 2022), to an increase of model resolution (Nicolsky et al., 2008; Aas et al., 2019; Aga et al., 2023). Latter allows to account for specific processes in detail, e.g. cryoturbation (Nicolsky et al., 2008) or ice segregation (Aga et al., 2023), but it also incurs high computational costs, making such resolution impractical in LSMs.

To bridge this gap, a common compromise is *tiling*, where grid cells are subdivided into tiles that interact with one another (Aas

et al., 2019; Cai et al., 2020). It enables the execution of a model on a coarser grid, which reduces computational demands, yet implicitly accounts for grid cell-internal heterogeneity and associated processes such as lateral fluxes of energy and water, as well as the movement of mass, e.g. redistribution of snow or soil particle displacement. In the context of permafrost-affected landscapes, reasonable tilings are characterised by the separation of the rim from the centre, or the plateau from the mire, within a polygonal tundra site (Aas et al., 2019; Martin et al., 2021). These tilings differ in terms of soil texture and related soil

properties, as well as in vegetation cover, height, and elevation. Alternatively, a classification based on soil ice content can be employed (Cai et al., 2020).

Despite tiling, and especially multi-scale tiling, where tiling works on different sub-grid scales, has been shown to improve the representation of permafrost-specific dynamics such as permafrost degradation (Aas et al., 2019; Nitzbon et al., 2021), it is still an approximation and lacks information about the actual spatial pattern of a certain landscape. For example, tiling

schemes cannot distinguish between coarse and fine patterns if the total proportion of properties within a grid cell is the same. This limitation is critical for accurately computing lateral fluxes, which occur at the transitions between contrasting zones. Furthermore, the highest tiling resolution used so far is -to our knowledge- about one meter (Nitzbon et al., 2021), still an order of magnitude coarser than the actual centimeter-scale heterogeneity of soil texture. Consequently, the potential of tiling to address the fundamental questions surrounding the influence of soil property aggregation within a grid cell on simulation

outcomes, and the role of actual spatial distribution of soil properties in this process, remains unanswered. Furthermore, the impact of resulting lateral fluxes on aggregation error, which are the foci of this study, remains to be elucidated.

Given the high *small-scale* heterogeneity and its effects on soil temperature dynamics in permafrost-affected ecosystems, we ask the following questions in this paper:

– What is the aggregation error in soil temperature due to sub-grid scale soil heterogeneity in non-sorted circles?

– What is the impact of heterogeneity-induced lateral heat fluxes on this aggregation error?

To address these questions, we apply a two-dimensional soil temperature model with high spatial resolution to simulate a non-sorted circle. We analyse how small-scale soil heterogeneity and heterogeneity-induced lateral heat fluxes affect surface and





subsurface temperatures, surface energy fluxes, snow depth, and active layer thickness. We discuss our findings in the context of land surface modeling in general and the simulation of permafrost-affected ecosystems in particular.

## 2  Methods

### 2.1  Soil Temperature Model

To address our research questions, we apply a standard vertical soil temperature model and extend its heat conduction scheme to a second spatial dimension. The third dimension is neglected to reduce computational cost and because high-resolution three-dimensional soil data are not available. To mitigate the effect of this simplification, we adopt a radially symmetric model
setup (see Section 2.2).

The vertical scheme of the model is based on well-known principles as described in Ekici et al. (2014) and Bonan (2019). In particular, it couples soil thermal and hydrological processes through (i) phase change of soil water (Ekici et al., 2014) and (ii) changes in soil thermal properties due to soil moisture (Bonan, 2019). The model also represents snow dynamics after Ekici et al. (2014). Building on this vertical framework, we extended the heat conduction scheme to a second (horizontal) dimension,
which is detailed described in the following subsection. The resulting two-dimensional, pedon-scale soil model—referred to as DynSoM-2D—simulates geophysical variables such as soil temperature, and soil water and ice content, as well as sensible and latent surface heat fluxes with sub-daily temporal resolution. The spatial resolution is user-defined: column widths are uniform, while layer thickness increases exponentially with depth in the standard configuration. Near-surface layers begin at centimeter scale and gradually increase to several meters at depth to ensure numerical stability.

DynSoM-2D requires information on geographic location, vegetation cover and soil texture, which can be recalculated at finer or coarser spatial resolution. The required soil texture information is the organic matter content and the fractions of clay and silt as part of the mineral soil, from which the sand fraction is calculated.

Soil input data are used to calculate site-specific parameters, such as soil porosity ($\Theta_{pore}$), through so-called *pedotransfer functions*, which estimate physical parameters, i.e. thermodynamic and hydrological parameters, from measurable soil proper-
ties. In the case of soil porosity, which is one of the *vanGenuchten parameters* (Van Genuchten, 1980), DynSoM-2D applies the pedotransfer functions from Wösten et al. (1999), which give values between 0.36 (silty clay) and 0.45 (silty loam). The estimated physical parameters are then used for further calculations.

In addition to this constant site information, DynSoM-2D requires a site-specific atmospheric forcing as an upper boundary condition.

120       In the following, we present the heat conduction scheme that is used in DynSoM-2D, and the calculation of the surface energy balance, with a special focus on their links to soil texture and its heterogeneity.





### 2.1.1 Heat Conduction Scheme

We assume that the soil temperature dynamics are governed by heat conduction and phase change. The classical one-dimensional heat conduction equation is extended to a second (horizontal) dimension (eq. 1). Instead of solving the two-dimensional heat conduction equation in an implicit scheme, we solve the vertical and horizontal components successively using a classical implicit one-dimensional finite difference method according to Bonan (2019). This approach is justified by the short time step of one hour (Bonan, 2019). In addition, the latent heat of fusion during phase change is considered following the approach of Ekici et al. (2014).

$$c_v \frac{\Delta T}{\Delta t} = \kappa_x \frac{\Delta^2 T}{\Delta x^2} + \kappa_z \frac{\Delta^2 T}{\Delta z^2} + \Delta T_{pc}, \; with \tag{1}$$

$$\Delta T_{pc} = E_{pc} \rho_i \frac{\Delta \Theta_i}{\Delta t} \tag{2}$$

with $\Delta T$ representing the change in soil temperature in time ($\Delta t$) or space ($\Delta z$ or $\Delta x$ respectively), as well as due to phase change ($\Delta T_{pc}$). $c_v$ is the volumetric heat capacity, and $\kappa_z$ and $\kappa_x$ are the heat conductivities in different directions. $\Delta T_{pc}$ is given by the specific latent heat for fusion ($E_{pc}$), as well as the change in ice content within the given time step ($\frac{\Delta \Theta_i}{\Delta t}$) and ice density ($\rho_i$).

At the bottom boundary, a zero-flux condition is assumed due to the large layer thickness and minimal thermal gradient. At the surface, a prescribed skin temperature (see following section) serves as the upper boundary condition. Also for the lateral boundaries we assume a zero flux and define the spatial domain accordingly (see sec. 2.2).

Soil temperature and heat fluxes are governed by the thermal conductivity ($\kappa$) and volumetric heat capacity ($c_v$), both of which depend on soil texture and composition. Typical values for various soil components are provided in table 1. To estimate the effective thermal conductivity for each grid cell, DynSoM-2D uses the Johansen method (Farouki, 1981), which interpolates between dry and saturated conductivity values. These endpoints, $\kappa_{\text{dry}}$ and $\kappa_{\text{pore}}$, are calculated as volumetric-weighted averages based on the thermal conductivities of the constituent materials. For $\kappa_{\text{dry}}$, this includes the mineral fractions (clay, silt, sand) and organic matter. For $\kappa_{\text{pore}}$, the pore-filling components—air, liquid water, and ice—are considered. The volumetric heat capacity for each grid cell is similarly computed as a weighted average of the component-specific heat capacities, using the volume fractions of each soil constituent.

### 2.1.2 Surface Energy Balance & Surface Temperature Calculation

The surface temperature, which is used as the Direchlet boundary condition for the soil temperature calculation, is determined by the energy exchange between the atmosphere and the soil, i.e. the surface energy balance (eq. 3). DynSoM-2D calculates the net radiation at the surface ($R_n$) from the incoming solar radiation ($S_r$), which is dependent on date, time and location, adjusted to the weather situation, and partly reflected by the surface albedo ($\alpha$), as well as the incoming and outgoing long wave radiation ($L_{in}, L_{out}$), which is determined by the temperatures, e.g. of the surface and clouds. The net radiation is balanced by





**Table 1.** Thermal properties for solid, liquid and ice components of soil originally taken from De Vries and Van Wijk (1963).

|  | Heat conductivity ($\kappa$) (W m$^{-1}$ K$^{-1}$) | Heat capacity ($c_v$) (MJ m$^{-3}$ K$^{-1}$) |
|---|---|---|
| Quartz | 8.80 | 2.12 |
| Clay minerals | 2.92 | 2.44 |
| Organic material | 0.25 | 2.50 |
| Liquid water | 0.57 | 4.19 |
| Ice | 2.18 | 1.88 |
| Air | 0.02 | 0.00 |

the sensible heat flux ($H$), the latent heat flux ($\lambda E$) and the soil heat flux ($G$).

$$R_n = (1-\alpha)S_r + L_{in} - L_{out} = H + \lambda E + G \tag{3}$$

Based on this balance, DynSoM-2D determines the heat in the soil using a skin layer approach (Bonan, 2019), which concen-
trates the energy balance between the atmosphere and the soil on a theoretical skin layer that is part of the soil profile. The heat
is recalculated to a skin layer/surface temperature, which is then used as an upper boundary condition for the soil temperature
calculation (eq. 1).

The surface energy balance is strongly influenced by vegetation and snow, or the presence of a pond, which are possible surface
covers in DynSoM-2D that change the surface albedo ($\alpha$) and thermal properties.

For snow, DynSoM-2D applies a minimum threshold of 3 cm for inclusion in energy and heat transfer calculations, following
Ekici et al. (2014). This threshold accounts for the patchy nature of early and late snow cover. Insulation effects depend on
total snow depth, which is discretised into 3 cm layers. Each layer is included in the vertical heat conduction calculations
once it is fully filled. Snow layers are treated analogously to soil layers for vertical conduction, but are excluded from the
lateral conduction scheme. This is based on the assumption that lateral temperature gradients in snow are minimal and that the
thermal conductivity of snow is very low. Albedo changes from bare soil albedo ($\alpha_{soil}$), which is depending on soil wetness, to
a constant snow albedo ($\alpha_{snow}$) as soon as the snow height exceeds the first 3cm. As albedo values vary with the wavelength
of incoming radiation, DynSoM-2D also applies different albedo values for radiation within the visible (VIS) and near-infrared
(NIR) spectrum, following Bonan (2019).

## 2.2 Model setup and protocol

For this theoretical study, we initialised soil texture and organic matter profiles based on a representative example of a non-
sorted circle described by Gentsch et al. (2015) (Supplement, profile CH-E), located near Cherskii, Siberia, Russia. We resam-
pled this data to a 10 cm × 10 cm grid over a spatial domain of 1 m in both vertical and horizontal directions. Horizontally,
the 1 m transect spans from the center of a non-sorted circle to its rim, allowing us to assume zero lateral fluxes at both bound-
aries—i.e., at the circle center and the midpoint to the adjacent circle. Vertically, the domain extends to a depth of 1 m, which



**Table 2.** Soil texture information for homogeneous soil (left) and heterogeneous soil (right). Values in brackets show shares of clay, silt, and sand from mineral soil only.

| | clay | silt | sand | OM | | | clay | silt | sand | OM |
|---|---|---|---|---|---|---|---|---|---|---|
| O-A-B | 11.34% | 34.65% | 17.01% | 37% | O | | 0.50% | 0.50% | 4.00% | 95% |
| | (18%) | (55%) | (27%) | | | | (10%) | (10%) | (80%) | |
| A-B | 16.40% | 54.70% | 8.20% | 18% | A | | 15% | 48.75% | 11.25% | 25% |
| | (20%) | (70%) | (10%) | | | | (20%) | (65%) | (15%) | |
| B | 17% | 59.50% | 8.50% | 15% | B | | 17% | 59.50% | 8.50% | 15% |
| | (20%) | (70%) | (10%) | | | | (20%) | (70%) | (10%) | |
| B-OA | 18.17% | 53.72% | 7.11% | 21% | OA | | 19.50% | 42.25% | 32.5% | 35% |
| | (23%) | (68%) | (9%) | | | | (30%) | (65%) | (5%) | |
| C | 14.25% | 76.00% | 4.75% | 5% | C | | 14.25% | 76.00% | 4.75% | 5% |
| | (15%) | (80%) | (5%) | | | | (15%) | (80%) | (5%) | |

reaches the parent rock layer at this site.

In addition, we averaged soil state variables laterally to obtain a reference homogeneous soil. The final set up is represented in figure 1 and tables 2.

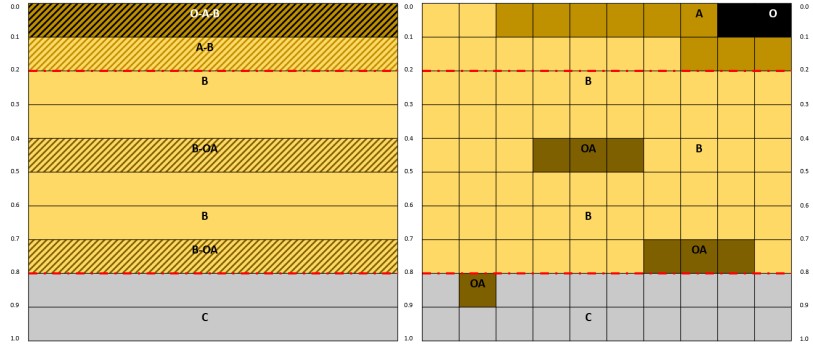

**Figure 1.** Schematic representation of soil input matrices for homogeneous soil (left), and heterogeneous soil (right). Values for clay, silt and sand, as well as for organic material can be found in table 2.

We excluded pond formation and vegetation growthto isolate the effects of soil texture heterogeneity. However, snow cover was included, given its essential role in permafrost energy dynamics. Atmospheric forcing was obtained from the CRUNCEP

Version 7 dataset (Viovy, 2018), using data specific to the study site.

We ran three different model simulations:

– **M0**—using the homogeneous soil input. This represents the standard one-dimensional soil model without pedon-scale heterogeneity and serves as the reference scenario.



- **M1**—using heterogeneous soil input, but assuming no lateral heat fluxes. The model structure remains one-dimensional but with fine-scale horizontal resolution.

- **M2**—using heterogeneous soil input with lateral heat transport enabled, thus extending the model to a fully two-dimensional setup.

To initialise the soil state variables, we performed a 50-year spin-up for each setup individually, repeating the 14-year atmospheric forcing. From the final run, we took the penultimate year for our analyses, because we needed three additional days from the years before and after the year being analysed to support the one-week running mean that we used to smooth our simulation results (unless otherwise stated). We chose to smooth our output in this way to avoid overly fuzzy near-surface results, which are strongly influenced by atmospheric forcing when the output is not smoothed, but to preserve the current atmospheric forcing signal, which is strongly superimposed when another averaging method is used, e.g. a 10-year mean.

Our analysis focuses on soil temperature differences among the three model configurations, specifically the effects of sub-grid soil heterogeneity. We distinguish between the *inter-model range*—i.e., differences in mean results across M0, M1, and M2—, which may be also presented by the absolute differences (eq. 4, $\Delta X$) between the heterogeneous soil models (M1, M2) and the homogeneous soil model (M0), and the *intra-model range*—i.e., spatial variation among columns in M1 and M2.

$$\Delta X = X^{het} - X^{hom} \tag{4}$$

## 3 Results

### 3.1 Soil parameters

Simulated soil porosity differs slightly between the homogeneous and heterogeneous soil setups, primarily due to the spatial variability of organic matter (OM) content in the latter (tab. 3). This variability has a pronounced effect on thermal properties. In general, the horizon-wise averaged heat conductivity ($\kappa$) is lower in the heterogeneous soil than in the homogeneous soil, because the relationship between heat conductivity and organic matter (OM) content ($\Theta_{OM}$) is not linear. OM-rich patches within the heterogeneous soil lower the mean heat conductivity more than an even distribution of OM within the homogeneous soil, even if the total amount of OM is the same. Conversely, the averaged heat capacity ($c_v$) is higher in the heterogeneous soil. This is linked to slightly increased porosity, which enhances total water content under saturated conditions. Since water has a higher heat capacity than mineral soil components, this leads to an overall increase in soil heat capacity. In contrast, the calculated surface albedo is hardly affected, and any changes are widely covered by the influence of soil moisture on albedo (fig. A1c&e).

### 3.2 Temporal Pattern

The surface heat flux to the atmosphere ($F_{surf}$), defined here as the sum of the sensible heat flux ($H$) and the latent heat flux ($\lambda E$, both see eq. 3) ranges from $-0.32 kW/m^2$ per day in winter (all models) to $0.45 kW/m^2$ (M0) to $0.51 kW/m^2$ (M2) per





**Table 3.** Simulated values for porosity ($\Theta_{pore}$) and resulting values for heat conductivity ($\kappa$), and heat capacity ($c_v$) for unfrozen, saturated soils. The values are spatially averaged for topsoil (0-20cm depth), subsoil (20-80cm depth), and deep soil (80-100cm depth) separately, as well as for the entire soil (0-100cm depth).

|  | $\Theta_{pore}^{hom}$ | $\kappa^{hom}$ | $c_v^{hom}$ | $\Theta_{pore}^{het}$ | $\kappa^{het}$ | $c_v^{het}$ |
|---|---|---|---|---|---|---|
| topsoil | 0.49 | 2.50 | 2.91 | 0.54 | 2.41 | 3.11 |
| subsoil | 0.49 | 2.60 | 3.02 | 0.53 | 3.14 | 3.23 |
| deep soil | 0.49 | 2.71 | 3.15 | 0.49 | 2.47 | 3.15 |
| soil | 0.49 | 2.60 | 3.02 | 0.50 | 2.65 | 3.08 |

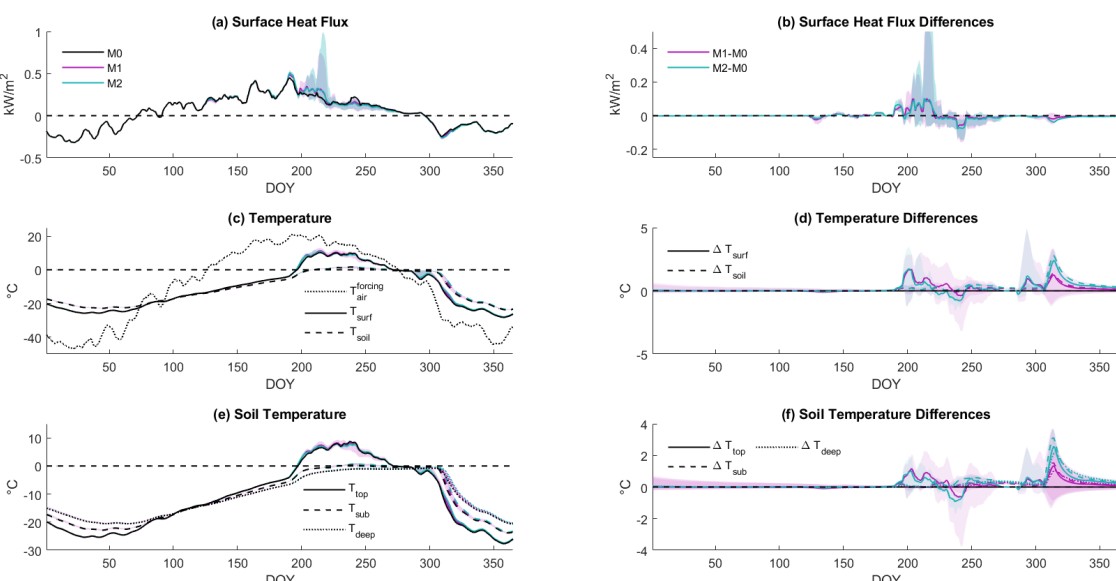

**Figure 2.** Simulation results (left; M0: black, M1: purple, M2: green) and deviations from reference model M0 (right; M1: purple, M2: green; eq. 4) for surface heat fluxes (top row), surface temperatures (middle rows), and soil temperatures (bottom row). Lines represent spatial mean, whereas shaded areas show the intra-model ranges, i.e. the ranges among columns for M1 and M2. (a) Simulated surface heat flux ($F_{surf}$), which is defined as sum of sensible heat flux ($H$) and latent heat flux ($\lambda E$). (b) Differences of surface heat fluxes ($\Delta F_{surf}$). (c) Simulated surface temperature ($T_{surf}$, solid lines), as well as soil temperature ($T_{soil}$, all depths, dashed lines), and atmospheric forcing temperature ($T_{atm}^{forcing}$, dotted line) for comparison. (d) Differences in surface temperature ($\Delta T_{surf}$, solid lines) and soil temperature ($\Delta T_{soil}$, all depths, dashed lines). (e) Simulated soil temperatures in topsoil (0-20cm depth; $T_{top}$, solid lines), subsoil (20-80cm depth; $T_{sub}$, dashed lines), and deep soil (80-100cm depth; $T_{deep}$, dotted lines). (f) Differences in topsoil ($\Delta T_{top}$, solid lines), subsoil ($\Delta T_{sub}$, dashed lines), and deep soil temperatures ($\Delta T_{deep}$, dotted lines).

day in summer (Fig. 2a). We find almost no differences between models within the first half of the year and also at the end

of the year. The intra-model ranges, i.e. the range among columns within the heterogeneous soil models M1 and M2, are also





small. Notable differences emerge only during summer and autumn when the soil is still snow-free. From July onwards, M0 emits significantly less heat on average than M1 and M2, which changes in mid-August until the mean curves converge again in September. This pattern corresponds to differences in (upper) soil moisture content, which influences surface albedo values (fig. A1). However, the inter-model range, i.e. the difference between the model means, does not exceed $0.1 kW/m^2$ per day,

indicating a difference of about 20% in heat fluxes to the atmosphere due to soil heterogeneity. In contrast, the intra-model ranges of heat fluxes increase to $0.6 kW/m^2$ (M1) and $0.9 kW/m^2$ per day (M2).

Despite these temporarily wider inter- and intra-model ranges, the total simulated annual heat budget differs by less than $1 kW/m^2$ between all model versions (M0: $13.7 kW/m^2$, M1: $14.1 kW/m^2$, M2: $13.5 kW/m^2$).

The simulated seasonal variations in surface and soil temperatures ($T_{surf}$, $T_{soil}$) across all model versions closely follow the

atmospheric temperature seasonality ($T_{atm}^{forcing}$) (fig. 2c, sec. 2.2), but the annual amplitude is reduced from 67.6°C ($T_{atm}^{forcing}$) to less than 40°C at the surface (M0: 38.5°C, M1: 39.4°C, M2: 38.8°C), and around 25°C in the soil (M0: 25.5°C, M1: 25.3°C, M2: 25.4°C).

Similar to the surface heat flux results, surface and soil temperature results do not differ much during the first half of the year, when the soil is frozen and snow-covered (fig. A1). From July to September, and again in November/December, the mean

temperature results deviate among models. During these periods, the homogeneous soil model M0 simulates a cooler surface than the heterogeneous soil models M1 and M2 (fig. 2c&d). However, during the last week of August (DOY 236-244), we observe a cooling effect due to soil heterogeneity, which is strong enough to affect the monthly mean temperatures of the surface and topsoil (tab. 4).

In general, the inter-model range of the simulated surface temperatures does not exceed 2.4°C per day, and is therefore smaller

than the individual intra-model ranges of M1 and M2 that increase up to 5.1°C (M1), and 4.9°C (M2) respectively. Interestingly the lateral fluxes, which lower the intra-model range in M2 compared to M1, increase the overall difference between M2 and M0 (annual mean: +0.25°C) compared to M1 and M0 slightly (annual mean: + 0.19°C; see also fig. 2d). We find a similar pattern for the soil temperature ranges, where either the inter-model range (maximum 2.8°C) and the intra-model range of M1 (4.5°C) are similar to the ranges of the surface temperature results, whereas the intra-model range of M2 is much smaller

(0.9°C). Both the inter- and intra-model ranges of simulated temperatures decrease when comparing monthly means, but are still present for annual means, which differ by 0.2°C ($T_{surf}$) and 0.2°C ($T_{soil}$), respectively (tab. 4).

Compared to the simulated surface temperatures ($T_{surf}$), the simulated soil temperatures ($T_{soil}$) differ only slightly, due to the fact that they are not only horizontally averaged over all ten columns, but also represent the entire upper soil down to 1 m depth (fig. 2c, tab. 4). We observe the annual cycle of temperatures at all depths, with the amplitudes decreasing with depth,

as the signal from the atmospheric forcing is attenuated by each soil layer above (fig. 2e&f). Simulated topsoil temperatures ($T_{top}$, 0-20cm depth) range from -27.7°C to 8.7°C, subsoil temperatures ($T_{sub}$, 20-80cm depth) range from -23.9°C to 0.8°C, and deep soil temperatures ($T_{deep}$, 80-100cm depth) range from -20.6°C to -0.6°C, with inter- and intra-model ranges varying between seasons.

We find only minor deviations between models (0.2°C on average until June) and a decreasing trend in the intra-model range

during the first half of the year, when the soil is still frozen (fig. A1e). However, the intra-model ranges for M1 are notable



**Table 4.** Monthly and annual mean of surface temperature and soil temperatures in different depths (topsoil: 0-20cm; subsoil: 20-80cm; deep soil: 80-100cm) in °C spatially averaged over all columns (upper row), as well as intra-model range (min/max; lower row), where present.

| | $T^{forcing}_{atm}$ | $T^{M0}_{surf}$ | $T^{M1}_{surf}$ | $T^{M2}_{surf}$ | $T^{M0}_{top}$ | $T^{M1}_{top}$ | $T^{M2}_{top}$ | $T^{M0}_{sub}$ | $T^{M1}_{sub}$ | $T^{M2}_{sub}$ | $T^{M0}_{deep}$ | $T^{M1}_{deep}$ | $T^{M2}_{deep}$ |
|---|---|---|---|---|---|---|---|---|---|---|---|---|---|
| Jan | -44.4 | -23.5 | -23.5 <br> -23.6–23.1 | -23.4 <br> -23.5–23.4 | -23.1 | -23.0 <br> -23.2–22.6 | -23.0 | -20.0 | -20.0 <br> -20.2–19.5 | -20.0 | -17.4 | -17.4 <br> -17.6–16.8 | -17.3 |
| Feb | -39.3 | -25.0 | -25.0 <br> -25.1–24.7 | -25.0 | -24.7 | -24.7 <br> -24.9–24.4 | -24.7 | -22.5 | -22.5 <br> -22.7–22.2 | -22.5 | -20.3 | -20.3 <br> -20.4–19.9 | -20.3 |
| Mar | -23.6 | -21.8 | -21.8 <br> -21.9–21.6 | -21.8 | -21.7 | -21.7 <br> -21.8–21.5 | -21.7 | -20.8 | -20.9 <br> -21.0–20.6 | -20.8 | -19.7 | -19.7 <br> -19.8–19.3 | -19.6 |
| Apr | -8.8 | -16.2 | -16.2 <br> -16.2–16.0 | -16.2 | -16.3 | -16.3 <br> -16.4–16.1 | -16.3 | -16.4 | -16.4 <br> -16.5–16.2 | -16.4 | -16.2 | -16.2 <br> -16.3–16.0 | -16.2 |
| May | -4.5 | -11.7 | -11.8 <br> -11.8–11.6 | -11.7 | -11.8 | -11.9 <br> -11.9–11.7 | -11.8 | -12.5 | -12.5 <br> -12.5–12.3 | -12.5 | -12.8 | -12.8 <br> -12.9–12.6 | -12.8 |
| Jun | 15.3 | -7.1 | -7.1 <br> -7.1–6.9 | -7.1 <br> -7.1–7.0 | -7.2 | -7.3 <br> -7.3–7.1 | -7.2 | -8.3 | -8.4 <br> -8.4–8.2 | -8.3 | -9.2 | -9.2 <br> -9.2–9.1 | -9.2 |
| Jul | 18.7 | 1.5 | 2.1 <br> 1.7–2.3 | 2.1 <br> 1.8–2.3 | 0.0 | 0.4 <br> 0.2–0.6 | 0.4 <br> 0.2–0.6 | -3.4 | -3.4 <br> -3.5–3.3 | -3.4 <br> -3.4–3.3 | -5.0 | -5.0 <br> -5.0–4.9 | -4.9 <br> -5.0–4.9 |
| Aug | 16.0 | 9.4 | 9.7 <br> 8.6–11.2 | 9.2 <br> 8.8–9.4 | 7.4 | 7.5 <br> 6.3–8.5 | 7.1 <br> 6.9–7.3 | -0.2 | -0.1 <br> -0.3–0.2 | 0.1 <br> 0.0–0.1 | -1.6 | -1.6 <br> -1.6–1.5 | -1.5 |
| Sep | 6.9 | 3.4 | 3.6 <br> 2.7–4.1 | 3.6 <br> 3.6–3.7 | 2.9 | 3.0 <br> 2.9–3.8 | 3.2 <br> 3.1–3.3 | -0.2 | 0.1 <br> -0.3–0.4 | 0.3 <br> 0.3–0.4 | -1.1 | -0.9 <br> -1.1–0.8 | -0.8 |
| Oct | -6.5 | -2.5 | -2.2 <br> -2.8–0.8 | -2.2 <br> -2.6–1.0 | -1.5 | -1.3 <br> -1.7–0.5 | -1.3 <br> -1.6–0.5 | -0.4 | -0.3 <br> -0.3–0.2 | -0.2 | -1.0 | -0.8 <br> -0.9–0.7 | -0.7 |
| Nov | -30.6 | -18.0 | -17.5 <br> -18.4–16.4 | -17.0 <br> -17.1–16.9 | -17.0 | -16.5 <br> -17.5–15.3 | -16.0 <br> -16.1–15.8 | -11.5 | -10.8 <br> -12.2–9.8 | -10.1 <br> -10.2–9.9 | -8.1 | -7.5 <br> -8.6–6.8 | -6.9 |
| Dec | -38.9 | -26.0 | -25.8 <br> -26.1–25.6 | -25.6 <br> -25.7–25.6 | -25.4 | -25.3 <br> -25.6–25.0 | -25.1 | -21.6 | -21.4 <br> -21.8–21.1 | -21.2 | -18.1 | -17.9 <br> -18.3–17.5 | -17.6 |
| mean | -10.9 | -11.5 | -11.3 <br> -11.7–10.8 | -11.3 <br> -11.4–11.1 | -11.5 | -11.4 <br> -11.8–10.9 | -11.4 | -11.5 | -11.4 <br> -11.6–11.1 | -11.3 <br> -11.3–11.2 | -10.9 | -10.8 <br> -11.0–10.5 | -10.7 <br> -10.7–10.6 |



(topsoil: 0.7°C, subsoil: 0.8°C, deep soil: 0.9°C), and are certainly greater than the intra-model ranges for M2 (topsoil: 0.0°C, subsoil: 0.1°C, deep soil: 0.0°C).

During summer, the inter-model ranges increase up to 1.2°C in the topsoil and 0.2°C in the subsoil, and the intra-model ranges also increase. Particularly striking is the intra-model range in the topsoil of M1, which exceeds 4°C on a daily basis and 2°C on a monthly basis at the simulated distance of 1m, and which is partly reflected in the relatively high inter-model ranges between M1 and M0 (daily: max. 1.2; monthly: max. 0.4°C) and M2 (daily: max. 1.0; monthly: max. 0.4°C).

After freezing, the simulated monthly mean temperatures differ by up to 1.5°C on average among all models across all depths, even on a monthly basis. However, the inter-model ranges are still mostly surpassed by the intra-model ranges. The spatial variation is around 2°C across all depths in M1 (topsoil: 2.2°C, subsoil: 2.4°C, deep soil: 1.8°C), but is significantly lower in M2. The inter-model range of M2 is only high in topsoil in October (1.0°C), but drops to below 0.5°C in November and December for both the subsoil and deep soil. Notably, M2 results are consistently warmer than M0 results, whereas the intra-model range of M1 and the respective differences show that some columns are warmer and some are colder than M0, but the mean is warmer (fig. 2f).

## 3.3 Spatial Pattern

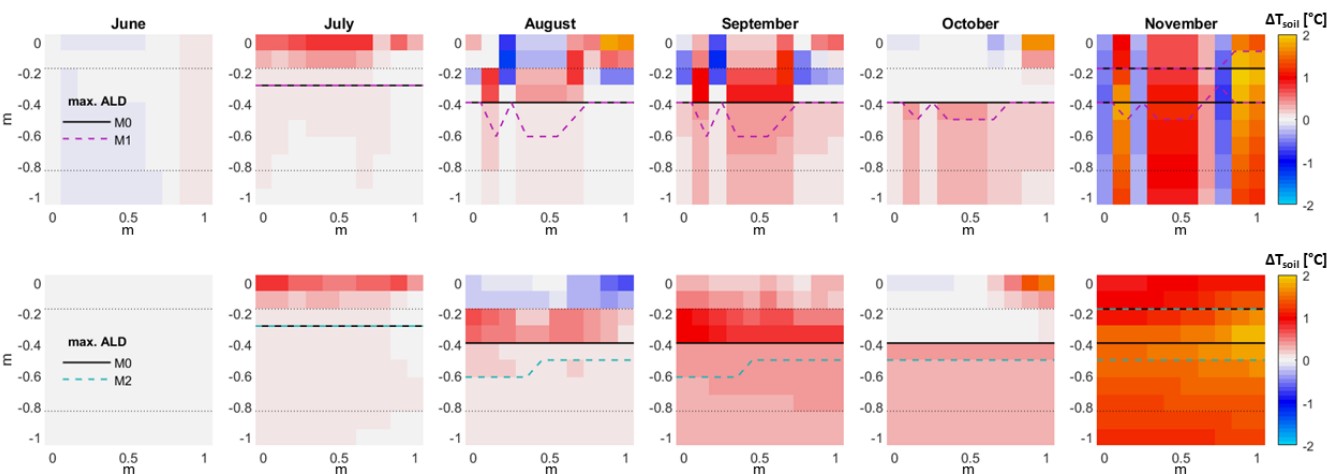

**Figure 3.** Mean differences in simulated soil temperature (eq. 4) per grid cell and max. active layer depth (ALD) for the months June-November. Upper row: Differences between M1 and M0. Lower row: Differences between M2 and M0. Colors represent differences in soil temperature. Thick lines show maximum active layer depth for M0 (solid, black) and M1/M2 (dashed, purple/green). Thin dotted lines separate topsoil from subsoil from deep soil. Full year data can be found in figure A2 in the appendix.

Simulated monthly mean temperatures in June do not differ significantly when comparing individual grid cells (fig. 3), which aligns with our temporal analysis since the soil is still frozen. Only two columns simulated by M1 are slightly warmer than M0, but this is balanced out by lateral fluxes in M2. Soil temperature differences begin to evolve in July, with the two



heterogeneous-soil models M1 and M2 simulating slightly warmer temperatures than the homogeneous-soil model M0 in the uppermost soil layers, which start thawing in mid-July (DOY 194). Thawing is uniform within all model variants and across all

columns, reaching 30 cm depth by the end of the month. We find that the maximum absolute warming of individual grid cells is equal in M1 and M2 compared to M0 (max. +0.8°C), but the affected area is larger in M2, where 27 grid cells are warmer than in M0 compared to 21 grid cells in M1. This leads to a minor but significant increase in average warming by 0.02°C across the entire domain compared to M0.

The spatial enlargement of heterogeneity-induced effects by lateral fluxes is even stronger in August. M1 shows a patchy

pattern of warming and cooling, especially in the topsoil, compared to M0, whereas M2's response is more laterally consistent, reflecting differences in soil moisture content that affect surface albedo (fig. A1). However, spatial mean temperatures in M0 and M1 are almost the same, whereas M2 simulates a cooler topsoil above a warmer subsoil (tab. 4, see also fig. 2f). Notably, the warmer patches in M1 compared to M0 are associated with grid cells that have a relatively high organic matter (OM) content (fig. 1),while cooler patches are associated with grid cells with a lower OM content. This is most evident at the surface,

but also in the deeper soil. Warming influences soil thawing and thus the thickness of the active layer. M0 simulates a maximum active layer thickness of 40 cm, while the active layer thickness varies between 40 cm and 60 cm in M1 and between 50 cm and 60 cm in M2. M1 simulates the deepest thawing in columns with deep OM-rich layers, and that this thawing is extended laterally in M2.

The temperature differences between M1 and M0 and M2 and M0 increase further in September, when the air temperature

is already cooling at the surface (fig. 2c and tab. 4). M1 simulates a column-wise patchy pattern of warming and cooling (max. ±1.2°C, +0.2°C on average) up to 40cm depth, and a warming of up to 0.4°C (+0.2°C on average) below, whereas M2 simulates a strong warming up to 1.0°C in the upper layers and a moderate warming (+0.3°C on average) below compared to M0 (fig. 3). Despite these larger differences in monthly mean temperatures, the maximum active layer depths remain the same in all model versions as in August.

Soil cooling continues in October, which leads to a retreat of the maximum active layer depth in M1 and M2 from a maximum of 60 cm to 50 cm. However, this is still 10cm deeper than the deepest thawing in M0. Besides, we find only minor differences between soil temperatures in topsoil, i.e. until 20cm depth, simulated by M0 and M1 or M2 respectively. Except of a warmer hot spot in the uppermost layer of column 9 and 10, where the OM-rich O-horizon is located (+1.7°C in M1, +1.5°C in M2). Below the topsoil, there is a 20 cm-thick layer where no temperature differences are observed among models or columns. This

is the so-called *zero curtain*, where soil temperature has reached 0°C and is freezing without further cooling until frozen. Since this happens in all simulations, temperature differences vanish. Below this freezing zone, M1 and M2 simulate a significantly warmer soil than M0 (M1: +0.2°C on average; M2: +0.3°C on average).

Soil freezing progresses until the soil is completely frozen after the first week of November, leading to the largest temperature differences between models on a monthly basis. During this week the soil freezes from both sides (fig. 3), i.e. from the surface,

which is strongly cooled by the atmosphere, and from below (fig. 2 and tab. 4). The M0 soil is already frozen after DOY 309, whereas M1 has a continuous unfrozen layer only until DOY 308, but remaining unfrozen cells until DOY 312, showing the uneven occurrence of cooling and freezing within a heterogeneous soil. On a monthly basis, M1 simulates soil temperatures




ranging from -0.7°C to +2.1°C compared to the corresponding M0 grid cells (+0.6°C on average). M2 has a continuous unfrozen layer until DOY 311, as laterally transported heat generally delays freezing, but then freezes completely within only

two days, as cooling is more uniform due to lateral fluxes. Nevertheless, M2 grid cells are up to 1.9°C warmer than M0 grid cells (average +1.4°C; fig. 3 and tab. 4).

### 3.4 Simulated soil temperature relation to soil texture and soil organic matter content

To explore the influence of soil texture in general and organic matter (OM) content ($\Theta_{OM}$) specifically on simulated soil temperatures in more detail, we aggregated all grid cells based on the information of the heterogeneous soil input data, i.e. O,

A, B, OA and C as shown in fig. 1, as well as based on their difference in OM content of the heterogeneous and homogeneous soil input data (tab. 2), and compare simulated soil temperature differences ($\Delta T_{soil}$, eq. 4).

We find that grid cells in the heterogeneous O-horizon simulate higher temperatures than the corresponding grid cells in the

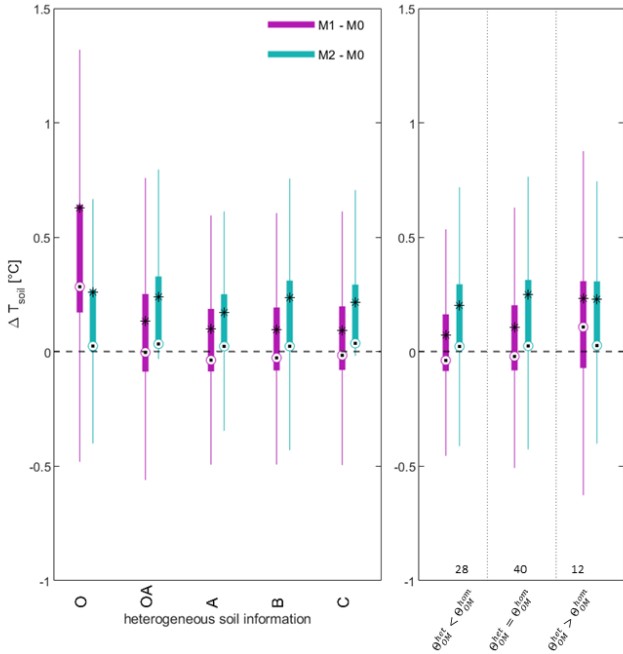

**Figure 4.** Differences in simulated daily soil temperature per grid cell between M1 and M0 (purple), as well as M2 and M0 (green) calculated by equation 4. Grid cells are aggregated by horizon classes of the heterogeneous soil input (left), as well as by differences in OM content ($\Theta_{OM}$) between the heterogeneous and the homogeneous soil inputs (right; values indicate the number of grid cells that fall in the respective category). Boxes represent quartile range of data and whiskers show the 0.1-0.9-quantile range. Circles show data median, whereas stars indicate data mean.

homogeneous soil (OAB horizon, see fig. 1). However, this relationship is weaker or not detectable for the other grid cells (fig. 4, left).



By aggregating the grid cells by their difference in OM content in the soil input, which can be higher in the heterogeneous soil compared to the homogeneous soil, lower, or equal, find that grid cells with a higher OM content tend to be warmer than grid cells with a lower OM content (fig. 4, right). This relationship is clearer when comparing simulations from M1 and M0, which show the direct effect of OM on soil temperature simulations, than when comparing simulations from M2 and M0, where lateral fluxes compensate for heterogeneity-induced differences between grid cells. In particular, for all data sets the median of

the differences is much lower than the mean, which is mostly in the upper quartile of the differences. This indicates that there are fewer days with very large differences that substantially increase the mean, and many days with small differences between simulated temperatures.

## 4    Discussion

We developed the high-resolution geophysical soil model DynSoM-2D and run the model at a typical non-sorted circle profile

in three different variants: with aggregated soil properties as reference (M0), considering heterogeneous soil, but without lateral fluxes (M1), and considering both heterogeneous soil and lateral heat fluxes (M2).

We analysed time series of simulated daily mean surface heat fluxes, surface and soil temperature, as well as temperature differences of monthly mean temperatures for each grid cell between these model setups. Additionally, we explored the effect of the homogeneous versus heterogeneous soil representation in general, and the impact of organic matter (OM) content on

soil temperature in particular.

Over the whole season, differences among models, i.e. the inter-model differences, are small when comparing mean annual values, and even smaller, when comparing median values. This is reasonable because all three model versions were run with the same forcing, with only the resolution of soil properties differing. In addition to small inter-model differences, we found comparatively large variations between columns of the heterogeneous setup, the so-called intra-model differences of M1 and

M2, which in most cases exceed the inter-model differences.

Generally, both, inter- and intra-model, differences are consistent with findings from previous studies, where differences of 0.5°C to 2°C (Beer, 2016), up to 2°C (Aas et al., 2019), or below 2.5C (Smith et al., 2022) are found between tiles, grid cells or pixels, respectively. Notably, those reported values are closer to our wider intra-model ranges, because they actually represent the differences between center and rim specifically or different pixels with varying soil parameters, which correspond to the

(outer edge) heterogeneous columns of our models M1 and M2, and thus to the intra-model ranges.

Differences in simulated temperatures and heat fluxes usually occur during summer, when the soil is snow-free and partly unfrozen. In general, soil heterogeneity has a warming effect in our simulations. Although we find temporarily wide deviations among simulated surface heat fluxes on a daily basis among models, but also across columns within models, the annual results differ little, because columns of the heterogeneous soil models that absorb most heat, also emit most heat (fig. 2a&b). Thus,

heterogeneity effects on opposite processes tend to cancel each other out to some extent.

The overall warming of the soil is slightly stronger and certainly more consistent across the columns when considering lateral fluxes. This shows that lateral fluxes can be important for the overall site-level soil temperature dynamics (figs. 2 & 3). Lateral



fluxes reduce the intra-model range of soil temperature and extend the intra-model range of heat fluxes (fig. 2b&d). This is due to lateral heat transport from surface grid cells that absorb most heat, or towards such grid cells that emit most heat, and explains, why the annual heat budget actually differs more between M0 and M1 than between M0 and M2, whereas the opposite is true for daily heat fluxes.

The difference in simulated temperatures is caused by differences in the thermal properties of the soil that alter the distribution of the heat flux (Bonan, 2019). We traced these differences back to the spatial distribution of organic matter (OM), which affects pore size distribution. Pore space can be either filled with air, which would lead to *negative* temperature changes in highly porous grid cells due to less heat storage, or water (or ice), which would lead to *positive* temperature changes in highly porous grid cells due to a much higher heat storage capacity. Our model tends to be very moist and close to saturation, which is why we observe a warming in OM-rich grid cells.

This link between OM and temperature changes is clearer when comparing M1 and M0 than when comparing M2 and M0 (fig. 4), because lateral fluxes in M2 redistribute heat among columns, thus offsetting heterogeneity-induced temperature differences. However, even for M1, we cannot find a clear relation between the relative difference of OM of a certain grid cell and the resulting difference in simulated temperature. This is due to the non-linear influence of OM on thermal properties, and possibly due to the impact of water content on thermal properties.

Neither for M2, nor for M1 we could detect a clear relationship between the soil input data, except for the O-horizon grid cells, and the change in simulated temperature for individual grid cells. This is, because the actual spatial location of a particular grid cell is important. For example, an A-horizon grid cell of heterogeneous soil will have a higher OM content when compared to an AB-horizon grid cell of homogeneous soil (e.g. 10-20 cm depth, see Fig. 1 and tab. 2 for comparison), but a lower OM content when compared to an (homogeneous soil) OAB horizon grid cell (0-10cm depth). In the first case, the A-horizon grid cell will simulate a warmer temperature than the corresponding AB-horizon grid cell, whereas in the second case it will simulate a cooler temperature than the corresponding OAB-horizon grid cell. This influence of the actual spatial location of certain grid cells, or rather the influence of the actual distribution of OM in the soil, questions the sufficiency of tiling approaches to account for sub-grid scale heterogeneity. Although tiling is a necessary step forward to account for sub-grid scale heterogeneity (Aas et al., 2019; Cai et al., 2020; Martin et al., 2021), the continued neglect of the actual spatial distribution of heterogeneous patches may reduce its utility.

Snow covers the surface and by that overlays any heterogeneity effects on surface albedo and insulates soil from the atmosphere, thus equalizing simulated surface heat fluxes. The only time when snow actually has an effect on simulated surface heat fluxes is at the end of the snow melt period. Snow melts unevenly on heterogeneous surfaces, which is why some columns of the heterogeneous soil are snow-free earlier than others, and also earlier (or later) than respective columns of the homogeneous soil. Since snow has a very different albedo from (bare) soil, the timing of a snow-free surface majorly affects surface heat fluxes. However, this effect is almost balanced out by the fact that only some columns of the heterogeneous soil show an earlier snow-free surface, while others are snow-covered longer than the homogeneous soil columns. The reason for the uneven snow melt is a change in the thermal properties of the (sub-)surface grid cells due to the heterogeneous soil textures. This change alters heat transport, which can either warm cells and lead to an earlier (and potentially faster) snow melt, or cool cells, leading



to a later snow melt (fig. 2a&b and A1a&b). Since land surface models underestimate the speed of snow melt and hence have soil temperature biases during the snow melt period Ekici et al. (2014), considering this effect of soil heterogeneity on snow melt could help improve future models.

Ice has a higher thermal conductivity than water (tab. 1). Therefore, heat is better transported through frozen soil than through unfrozen soil. This happens throughout the year in the deeper, perennially frozen layers, which is why the temperature of the deep soil layers differs least between the M1 and M2 columns, and also between the models. However, it also causes the differences in simulated temperatures of upper layers, which thaw during the summer, to disappear (fig. 2e&f). This is particularly noticeable in early winter, when large differences due to uneven freezing disappear from November to January (fig. 3 and fig. A2). As the soil cools, the grid cells reach the freezing temperature of 0°C at slightly different times due to their previous temperature differences and their specific heat capacity and conductivity, both of which depend on the soil texture. They then begin to freeze, whereby the duration of freezing depends (i) on the required energy for phase change, and (ii) on the allowance of additional lateral heat fluxes. The latter delays the time of initial freezing within a given soil layer, as heat is primarily transported between the columns until all have cooled to 0°C before the first grid cell freezes, but also shortens the duration of the layer's freezing, as all the grid cells have reached freezing temperature by the time the first cell freezes.

Due to the high heat release of freezing and the resulting time shift of further cooling, temperature differences that evolved during summer are intensified and lead to very high temperature ranges after freezing. Notably, we do not see a similar increase of deviations during thawing, because the temperature differences before thawing are small, and heat is transported comparably fast through frozen soil, so that the soil thaws relatively even among columns, but also among models. Thus, thawing happens rather even, whereas freezing happens very uneven within the heterogeneous soil and builds frozen patches that turn into cooler spots.

In terms of an aggregation error, our findings demonstrate that there is a notable shift towards lower soil temperatures when neglecting sub-grid scale soil heterogeneity and the actual spatial distribution of soil properties, which cannot be resolved by tiling. We admit that the shift towards *higher* temperatures when considering small-scale heterogeneity is caused by a high soil moisture, and that simulations with lower soil moisture may show the opposite, but the temperature shift, i.e. the aggregation error, itself is substantial. In addition, the temperature change is larger when comparing M2 and M0 than when comparing M1 and M0, showing an amplification of the error when considering heterogeneity-induced lateral fluxes in the small-scale model, which will be consistent for any aggregation error. Notably, the detected aggregation error does not affect the annual surface heat budget much, meaning that the heterogeneous soil models simulate a warmer soil without absorbing (or emitting) significantly more energy on annual basis. This finding may reduce the impact of the detected aggregation error on regional or global simulations, as it propagates only marginally into the atmosphere on an annual basis.

However, and although a warming of 0.23°C on average between M0 and M2 (+0.11°C on average between M1 and M0) seems minor at first sight, it has great effects on this specific permafrost-affected site by deepening soil thawing and speeding up snow melt, which may impact the dynamics of the entire ecosystem largely (Chadburn et al., 2015b, a; Beer, 2016; Beer et al., 2018; Aas et al., 2019; Burke et al., 2020; Rixen et al., 2022; Schädel et al., 2024). In addition, the bias of individual grid cells up to 5°C in summer will affect temperature-dependent soil processes such as heterotrophic respiration. Since the



temperature dependency of biochemical reactions is highly non-linear (Langridge, 1963; Conant et al., 2011; Bonan, 2019), ultimate effects can be even more amplified and call for more studies including carbon (C) and nutrient cycles.

Simulated soil thaws up to 20cm deeper in summer when accounting for pedon-scale heterogeneity, which is in line with previous studies (Beer, 2016; Nitzbon et al., 2021), and also the duration of thawing is extended by three days (M1) or four days (M2) in fall (figs. 2 & 3 & A1). Since the faster snow melt extends the snow-free time by up to five days in spring, snow melt and thawing effects may together prolong the simulated vegetation growing season by more than one week, when accounting for sub-grid scale heterogeneity in land-surface models. Additionally, the deeper thawing may allow plants to root

deeper and liberate nutrients, which enhances plant growth, as well as it may promote decomposition of deeper soil organic matter. All these effects impact the simulations of ecosystem dynamics and thus may affect its simulated C balance significantly in consequence of the aggregation error due to neglecting sub-grid scale soil heterogeneity (Burke et al., 2012, 2020; Schädel et al., 2024).

   Although we included lateral heat fluxes, we did not include lateral water fluxes in the model yet, which affect soil temper-

ature directly through heat advection, but also indirectly due to redistribution of water and ice that largely influence thermal properties (Cai et al., 2020).

   The neglect of heat advection (Gao and Coon, 2022) directly affects soil temperature because heat advection may be as strong or even stronger than heat conduction (Nicolsky et al., 2008; Bonan, 2019). In case heat conduction and heat advection work in the same direction, they would enhance their effects and by that mitigate soil temperature differences among columns faster.

But in case they flow in opposite directions, they would attenuate the effect of each other and thus potentially intensify temperature differences among columns (Nicolsky et al., 2008; Aas et al., 2019).

   Regarding the build of ground ice lateral water fluxes may reinforce the spatial occurrence of ice by flowing towards the freezing front and thus delivering more water that can freeze, or mitigate it, because the delivered water may transport heat towards the freezing front (Nicolsky et al., 2008; Aas et al., 2019; Fu et al., 2022).

Using only one test site reduces our ability to upscale our results to regional or even global scales. We cannot be certain that soil warming due to sub-grid scale soil heterogeneity will remain, if the experiment is replicated at another site or if the spatial resolution is increased. Also, the potential effects on ecosystem dynamics and the associated C budget may vary between sites and change with scale. However, the fact that lateral fluxes actually amplify, rather than attenuate, the aggregation error of sub-grid scale soil heterogeneity in terms of soil temperature gives us an indication that the heterogeneity-induced soil temperature

change, here a warming, as well as the potential effects on ecosystem dynamics, are likely to be consistent and remain when simulating other sites. It may also persist when the resolution is increased, but since we found that the actual distribution of heterogeneous variables such as OM content plays an important role, the risk of underestimating the aggregation error is high. Also, tiling may only partially account for this aggregation error by neglecting the true distribution of state variables.



## 5 Conclusion

Simulation experiments with the high-resolution soil model DynSoM-2D show a notable - in our study negative - aggregation error in soil temperature, which affects the simulated depth of the active layer and the duration of the snow-free period in summer. Since the general temperature shift is more pronounced and more consistent when considering heterogeneity-induced lateral heat fluxes, we assume that it will remain at other sites and under changed environmental conditions, although the direction of the aggregation error may differ. Temperature changes are most pronounced in the near-surface soil, which make them

especially relevant for ecosystem dynamics, but disappear with depth and appear to be less relevant for energy exchange with the atmosphere.

   We traced the detected temperature shift back to the actual spatial distribution of organic matter, which nonlinearly alters soil thermal and hydrological properties. Due to this non-linear relationships and the importance of the actual organic matter distribution, current approaches to account for sub-grid scale heterogeneity, such as tiling, may still underestimate the aggregation

error of soil heterogeneity and its effect on ecosystem dynamics. Thus, further studies are needed to better quantify these effects and estimate their importance on regional to global scale.

*Code availability.* Code and necessary forcing data are available under https://doi.org/10.25592/uhhfdm.17610 (Thurner et al., 2025).



## Appendix A: Results - Additional Plots

For better presenting deviations between model simulations, we do not only analyse actual simulation results, but also absolute
differences (eq. 4, $\Delta X$), as well as relative changes (eq. A1, $\delta Y$) between the heterogeneous soil models (M1, M2) and the
homogeneous soil model (M0).

$$\delta Y = \frac{Y^{het} - Y^{hom}}{Y^{hom}} \tag{A1}$$

### A1    Temporal Pattern

The difference in soil porosity due to the difference between the homogeneous and the heterogeneous soils (tab. 3) also impacts
the soil water and ice contents (fig. A1). Although the inter-model range is small, the intra-model ranges are wide, especially
at the topsoil, where the heterogeneity is most pronounced, and the amount of organic material (OM), which has an extremely
high porosity, differs most. In contrast to soil temperature simulations, lateral heat fluxes do not affect the intra-model ranges
here, but they impact the inter-model range, i.e. the deviation of M2 from M0 compared to M1 to M0, because the warmer soil
of M2 contains more liquid water and less ice during the summer.



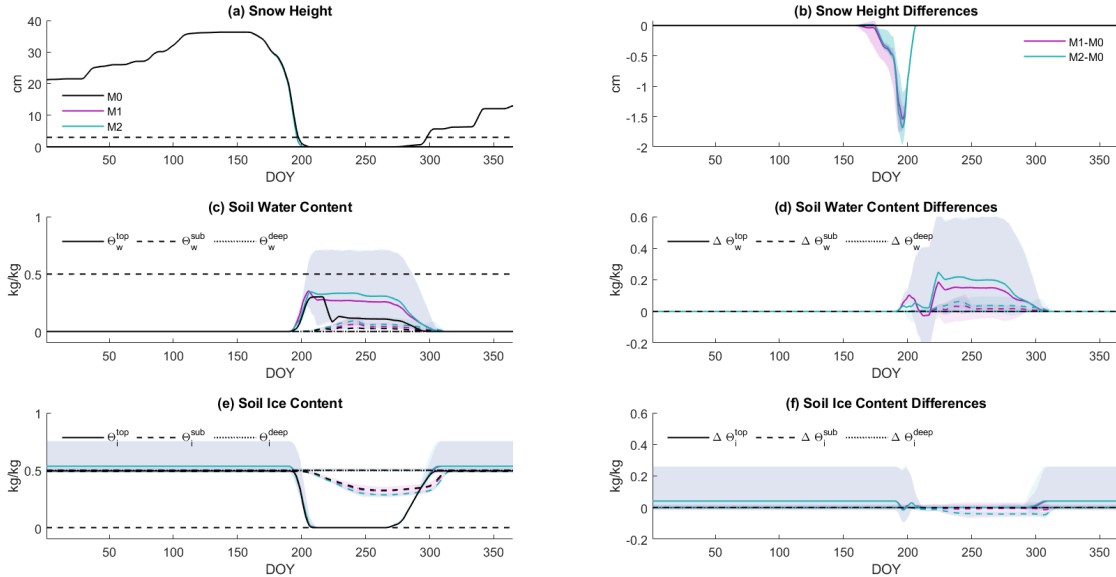

**Figure A1.** Simulation results (left; M0: black, M1: purple, M2: green) and deviations from reference model M0 (right; M1: purple, M2: green; eq. 4) for snow height (top row), soil water content (middle rows), and soil ice content (bottom row). Lines represent spatial mean, whereas shaded areas show the intra-model ranges, i.e. the ranges among columns for M1 and M2. (a) Simulated snow height. Dashed line indicates 3cm-threshold for snow effects on surface albedo and soil insulation. (b) Differences in simulated snow height. (c) Simulated soil water content in topsoil (0-20cm depth; $\Theta_w^{top}$, solid lines), subsoil (20-80cm depth; $\Theta_w^{sub}$, dashed lines), and deep soil (80-100cm depth; $\Theta_w^{deep}$, dotted lines). (d) Differences in topsoil ($\Delta\Theta_w^{top}$, solid lines), subsoil ($\Delta\Theta_w^{sub}$, dashed lines), and deep soil water contents ($\Delta\Theta_w^{deep}$, dotted lines). Black dashed line represents mean porosity ($\Theta_{pore}^{soil}$, tab. 3) as upper limit for $\Theta_w$. (e) Simulated soil ice content in topsoil (0-20cm depth; $\Theta_i^{top}$, solid lines), subsoil (20-80cm depth; $\Theta_i^{sub}$, dashed lines), and deep soil (80-100cm depth; $\Theta_i^{deep}$, dotted lines). Black dashed line represents mean porosity ($\Theta_{pore}^{soil}$, tab. 3) as upper limit for $\Theta_i$. (f) Differences in topsoil ($\Delta\Theta_i^{top}$, solid lines), subsoil ($\Delta\Theta_i^{sub}$, dashed lines), and deep soil ice contents ($\Delta\Theta_i^{deep}$, dotted lines).



## 475 A2 Spatial Pattern

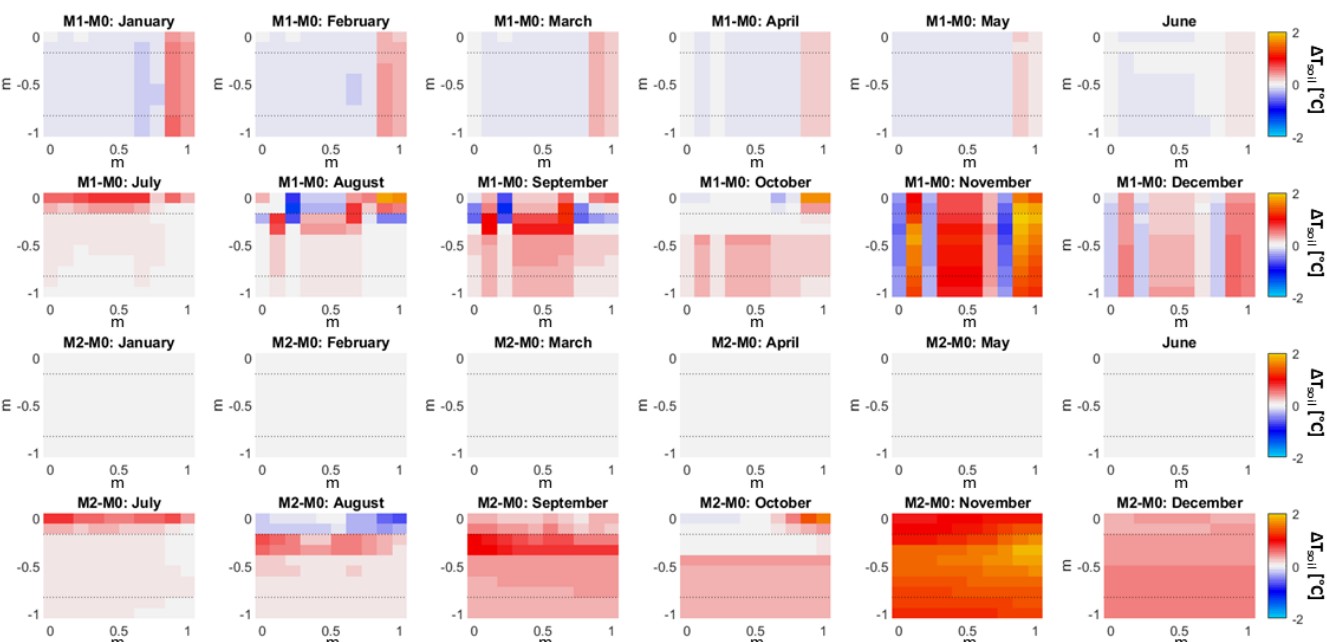

**Figure A2.** Mean differences in simulated soil temperature per grid cell (eq. 4). Upper row: Differences between M1 and M0. Lower row: Differences between M2 and M0. Colors represent differences in soil temperature. Thin dotted lines separate topsoil from subsoil from deep soil.





## A3 Relation to organic material content

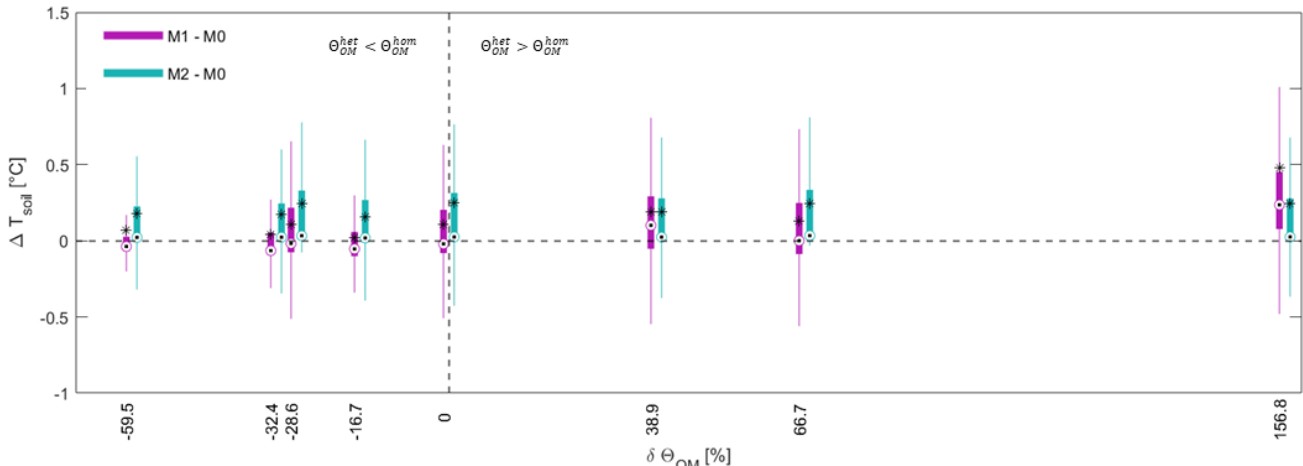

**Figure A3.** Differences in simulated daily soil temperature per grid cell between M1 and M0 (purple), as well as M2 and M0 (green) calculated by equation 4. Grid cells are aggregated by changes in OM content ($\delta\Theta_{OM}$, eq. A1) between the heterogeneous and the homogeneous soil inputs. Boxes represent quartile range of data and whiskers show the 0.1-0.9-quantile range. Circles show data median, whereas stars indicate data mean.



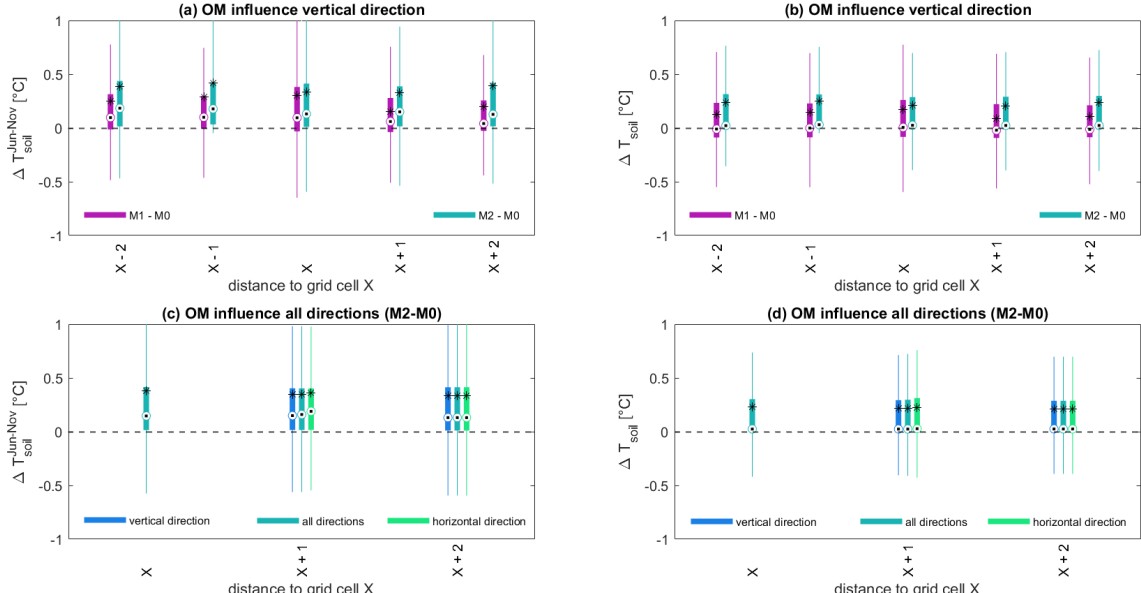

**Figure A4.** Differences in simulated daily soil temperature per grid cell (eq. 4). Values are aggregated by their position relative to OM-richer grid cells (eq. A1). Boxes represent quartile range of data and whiskers show the 0.1-0.9-quantile range. Circles show data median, whereas stars indicate data mean.

Upper row: Differences between M1 and M0 (purple), as well as M2 and M0 (green), whereby only vertical direction is taken into account, but separated into above and below the OM-richer grid cell. (a) Only data from June to November. (b) Full year data.

Lower row: Differences between M2 and M0. OM-influence is tested in vertical direction (blue), horizontal direction (light green), and both direction in combination (green). No differentiation between above/below or left/right neighboring. (c) Only data from June to November. (d) Full year data.



*Author contributions.* MT and CB designed the study. MT, XR and CB developed the model. MT performed the analyses, interpreted the results and wrote the initial manuscript. CB and XR contributed to writing the manuscript.

*Competing interests.* The authors declare that there are no competing interests.

*Acknowledgements.* MT and CB acknowledge funding by the Deutsche Forschungsgemeinschaft (DFG BE 6485/1-1 and BE 6485/2-1, BE 6485/2-2). XR acknowledges funding from the German Ministry for Education and Research (BMBF 03F0931A). We would like to acknowledge the use of OpenAI's ChatGPT and LanguageAI's DeepL for editing the mannuscript with regard to wording and grammar. After using, the authors reviewed and edited the content as needed and take full responsibility for the content of the submitted manuscript.





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
