# Peer review of "Lateral heat fluxes amplify the aggregation error of soil temperature in non-sorted circles"

_EGUsphere, 2025_

## Referee Comment (RC1)

**General assessment**

This study uses a pedon-scale soil physical model - called DynSoM-2D – to describe how small-scale heterogeneity affects soil thermodynamics and surface heat fluxes in a specific type of permafrost topography (non-sorted circles), with a particular focus on the effect of heterogeneity-induced lateral heat fluxes. This is an important topic, since most land surface models fail to represent subgrid-scale heterogeneity in permafrost-affected areas, thereby leading to an inaccurate representation of permafrost dynamics and biogeochemistry. The approach, tools and simulation design employed in this study are highly relevant for addressing these questions. The discussion is particularly interesting and provides an insightful interpretation of the results. The code is clear and well commented, making it easy to read.

Unfortunately, there are some major problems. First, the quantitative analysis is based on a single year, which is insufficient for deriving robust conclusions about the effect of soil heterogeneity on soil temperature. While the authors briefly explain their decision to conduct a single-year analysis (l.193), the results are highly dependent on the specific atmospheric conditions of that year. More robust results could be obtained by including several years in the analysis (e.g. a 10-year climatology or a longer timeseries), especially given that this modelling study is not limited by a lack of data.

Second, (inter- and intra-) model ranges are highly sensitive to extreme values and are not absolute metrics. They lack a reference point, such as a mean or median, against which they can be compared (for example, an increase in the intra-model range does not have the same impact on permafrost physics and biogeochemistry at high temperatures or close to 0°C). Although extreme values can be informative, other statistical metrics (e.g. mean/standard deviation or median/interquartile range) are a more robust choice for describing the results.

Finally, as this is a new model that has not been described elsewhere, a much more detailed description is required, particularly with regard to snow, hydrology and the calculation of soil thermal properties. In particular, the description lacks the key equations that underpin the results presented in this study. Some specific points are detailed in the "Specific Comments".

Overall, this is an interesting study that contributes to a growing body of literature on an important and challenging topic. It would definitely be a valuable contribution for the land surface modelling community. However, major revisions are required in a number of areas.

**Specific Comments**

- The title is quite technical, making it difficult to guess what the article is about at first glance (for instance, it does not mention "soil heterogeneity"). I do not have any other suggestions, but the article would benefit from a clearer title.
- l.101 : The principles to which the authors are referring to are unclear. Maybe just keep the following sentence (l.102) or combine both ?
- l.108-109 : Could you give the soil layer thickness for all layers ?
- l.111-112 : Bulk density is missing from the list (needed in pedotransfer functions of Wösten at al. (1999)).
- l.115 : What about the other soil parameters besides soil porosity ?

- l.118 :  The variables driving the model should be specified. Also specify how rainfall and snowfall are separated from total precipitation.
- l.140 : Please provide the equations used to calculate soil thermal conductivity and capacity as these parameters largely control soil temperature dynamics.
- Given the crucial role of soil water content and frozen/unfrozen fractions on soil temperature, the soil hydrological scheme should be described (including boundary conditions).
- l.156 "The heat is recalculated to a skin layer/surface temperature" is not very precise and should be clarified.
- l.160-169 : The equations governing snow cover dynamics should be provided, along with the values of snow conductivity, capacity and any other parameters controlling snow dynamics.
- l.174 : For clarity, please recall that the soil column extends below 1m. If I have understood the code correctly, the soil column is 50m deep, but this should be clearly explained in the manuscript.
- l.175 : Which soil properties are used at depths below 1m ?
- Fig.1 : The vertical axis needs a unit/label.
- l.178 : Vegetation growth is excluded but it is unclear whether vegetation is present in the simulations. If so, please describe the type of vegetation present.
- Table 2 : The percentages on the left do not add up to 100% for A-B.
- l.188 : The authors should provide evidence that the soil temperatures and soil moisture are close to equilibrium after the 50-year spinup (with no long-term drift).
- l.191-193 : Analysing a single year does not produce robust quantitative results, as these depend on the atmospheric forcing of that particular year (e.g. the amount of precipitation influencing both snow cover and soil thermal properties via soil moisture). The aggregation error could be different if a different year had been chosen. Why not use a longer time period ? I do not understand the need to "preserve the current atmospheric signal" (l.192). The results would be more robust if a longer period was included in the analysis, for example daily or monthly averages over 10 years, or the analysis of a longer timeseries if the authors prefer not to use time averages.
- l.201 : Please mention how $\theta^{het}$ is calculated. Horizon-wise average ?
- l.203 "In general, the horizon-wise averaged heat conductivity is lower in the heterogeneous soil than in the homogeneous soil" → This statement is not supported by the data in Table 3 where $\kappa^{het} > \kappa^{hom}$ for both the subsoil and the total soil.
- l.208-210 : I do not think it can be concluded from Fig.A1c&e that "the calculated albedo is hardly affected". The authors should provide data to support this statement.
- l.234 : What does "per day" mean ? I suppose it refers to daily values but this should be clarified.
- l.255 : "at the simulated distance of 1m" → unclear what this refers to.
- l.257 : "After freezing" → unclear, please precise.
- Using DOY in Fig.2 and months in Fig.3 makes it difficult to compare the two figures. Adding months to Fig.2 would improve understanding.
- l.276 : "reflecting differences in soil moisture content that affect surface albedo" → albedo data are missing to support this statement.
- l.300 :  What does "from below" refer to ? The freezing front ?
- Fig.4 : A statistical test, such as a t-test, is required in order to assess the significance of soil temperature differences.
- Fig.4 : The difference in soil temperature is likely to be dominated by snow and ice in winter, resulting in values close to zero from February to June. Maybe focusing on the period when soil temperature differences are significant (July to January) would help to assess the effect of

OM distribution on these differences, and would avoid pulling the median down artificially due to the near-zero $\Delta T_{soil}$ during half of the year ?

- Section 3.4 lacks quantification. Statistical metrics and tests should be used.
- l.315-317 : "By aggregating...lower OM content" → not sure if this is significant. A statistical test is needed.
- l.317-318 : "this relationship...from M2 to M0" → based on which metrics ?
- l.356-357 : This is counter-intuitive as OM generally insulates the ground against warmer air temperatures in permafrost regions (Zhu 2019, Loranty 2018). However this can be explained by the higher thermal conductivity in columns with a higher OM content, due to higher soil moisture, as the authors explain. A figure showing the 2D evolution of soil moisture (vertical and horizontal, similar to Fig.3) would support this statement.
- l.375-376 : "The only time when snow actually has an effect on simulated surface heat fluxes is at the end of the snow melt period" → rather on the difference of surface heat fluxes between homogeneous and heterogeneous configurations than on surface heat fluxes themselves (which are always impacted by snow) ?
- l.409 : "which will be consistent for any aggregation error." → I do not think this conclusion can be drawn from a single-site study. Such a conclusion would require simulations for other sites, as the amplification of the aggregation error by lateral fluxes probably depends on the spatial distribution of soil texture and OM content.
- l.443-446 : I do not understand why the amplification of the aggregation error by lateral fluxes provides more confidence that these results would remain for other sites.
- l.448 : In the context of land surface modelling, there are no global maps of soil texture and OM at a 10 cm resolution. Could you briefly describe how these results could help improving land surface models ?
- l.453 : "we assume that it will remain at other sites and under changed environmental conditions" → I do not think this assumption can be made based on a single-site study.
- Fig. A3 and Fig.A4 are not referenced in the main text. They should either be included or removed.

**Technical corrections**

- Bonan 2019 is cited for various aspects of the model and it would be helpful to provide more specific references for each citation (e.g. refer to the relevant chapter in the book).
- l.105 : Please give the section number.
- l.123 : "phase change" → water phase change
- l.126 : Please refer to the specific chapter/equations from Bonan (2019).
- l.131-134 : Please precise the units.
- l.136 : Please give the section number.
- l.138 : Please precise the units.
- l.154 : Please refer to the specific chapter/equations from Bonan (2019).
- l.171 : Please specify which figure from Gentsch et al. (2015) you are referring to.
- Table 2 : Please define O, A, B and C in the table legend.
- Figure 1 : Please precise in the legend what the red dash-dot lines refer to (separation topsoil, subsoil and deep soil ?).
- Table 3 : Please add units.
- Fig.2 (b) : The peak of maximum difference looks cropped.
- l.234 : "In general" should be removed.
- deVrese et al. Paper is now published and is no longer a preprint.

**References**

Loranty, M. M., Abbott, B. W., Blok, D., Douglas, T. A., Epstein, H. E., Forbes, B. C., Jones, B. M., Kholodov, A. L., Kropp, H., Malhotra, A., Mamet, S. D., Myers-Smith, I. H., Natali, S. M., O'Donnell, J. A., Phoenix, G. K., Rocha, A. V., Sonnentag, O., Tape, K. D., and Walker, D. A.: Reviews and syntheses: Changing ecosystem influences on soil thermal regimes in northern high-latitude permafrost regions, Biogeosciences, 15, 5287–5313, https://doi.org/10.5194/bg-15-5287-2018, 2018.

Wösten, J., Lilly, A., Nemes, A., and Le Bas, C.: Development and use of a database of hydraulic properties of European soils, Geoderma, 90, 169–185, 1999.

Zhu, D., Ciais, P., Krinner, G. *et al.* Controls of soil organic matter on soil thermal dynamics in the northern high latitudes. *Nat Commun* **10**, 3172 (2019). https://doi.org/10.1038/s41467-019-11103-1

---

## Author Comment (AC3)

**General assessment**

This study uses a pedon-scale soil physical model - called DynSoM-2D — to describe how small-scale heterogeneity affects soil thermodynamics and surface heat fluxes in a specific type of permafrost topography (non-sorted circles), with a particular focus on the effect of heterogeneity-induced lateral heat fluxes. This is an important topic, since most land surface models fail to represent subgrid-scale heterogeneity in permafrost-affected areas, thereby leading to an inaccurate representation of permafrost dynamics and biogeochemistry. The approach, tools and simulation design employed in this study are highly relevant for addressing these questions. The discussion is particularly interesting and provides an insightful interpretation of the results. The code is clear and well commented, making it easy to read. Unfortunately, there are some major problems.

We are grateful to the reviewer for emphasizing the importance of this study and the constructive feedback. We will address the mentioned problems point by point.

First, the quantitative analysis is based on a single year, which is insufficient for deriving robust conclusions about the effect of soil heterogeneity on soil temperature. While the authors briefly explain their decision to conduct a single-year analysis (I.193), the results are highly dependent on the specific atmospheric conditions of that year. More robust results could be obtained by including several years in the analysis (e.g. a 10-year climatology or a longer timeseries), especially given that this modelling study is not limited by a lack of data.

We agree with the reviewer that a single year data is not enough to describe the soil temperature of a site. However, the aim of our paper is to identify the differences on the temperature induced by heterogeneity of the soil and related lateral heat fluxes and avoid errors linked to aggregation. As stated on the manuscript, the simulations are preceded by a 50 year spin up that provides the conditions for the final run with aims to be a "snap-shot" of the differences on the soil. As the specific atmospheric conditions are exactly the same for all the soil columns, the differences captured on the "snap-shot" are solely due the soil heterogeneity. The inclusion of the average of several years to the analysis creates an out-averaging of effects that smooths out the differences between the different columns of the soil, dampening the differences in a similar way than the spatial aggregation does and masking the heterogeneity effect. In fact, we used the time average smoothing effect to smooth the output to avoid the overly-fuzzy results on the surface (as stated on the lines 191-192 of the manuscript). Nevertheless, we will add figures (similar to fig. 2) for other exemplary years to the SI to show that the general pattern (difference due to heterogeneity is enlarged by lateral fluxes) does not depend on the chosen year.

Second, (inter- and intra-) model ranges are highly sensitive to extreme values and are not absolute metrics. They lack a reference point, such as a mean or median, against which they can be compared (for example, an increase in the intra-model range does not have the same impact on permafrost physics and biogeochemistry at high temperatures or close to 0°C). Although extreme values can be informative, other statistical metrics (e.g. mean/standard deviation or median/interquartile range) are a more robust choice for describing the results.

We assume that the reviewer might have misunderstood, what we have done, and in that case, we will clarify this in the manuscript. We have used mean simulations among columns of the simulations MO - M2 to derive the inter-model differences, meaning that the inter-model difference between M1 and M0 is the difference between the mean over all columns of M0 and M1, respectively. The intra-model differences refer to the largest differences between columns of the same model, which is only applicable for M1 and M2, because M0 has a homogeneous soil input and consequently no differences among columns. These ranges (for M1 and M2) are highlighted as shadowed areas in our timeseries figures and include the minimum and maximum range of simulations.

Finally, as this is a new model that has not been described elsewhere, a much more detailed description is required, particularly with regard to snow, hydrology and the calculation of soil thermal properties. In particular, the description lacks the key equations that underpin the results presented in this study. Some specific points are detailed in the "Specific Comments".

We will add the specific chapter and/or equation from Bonan (2019), and add Beer et al. (2018) as reference fir our snow model, where necessary.

Overall, this is an interesting study that contributes to a growing body of literature on an important and challenging topic. It would definitely be a valuable contribution for the land surface modelling community. However, major revisions are required in a number of areas.

**Specific Comments**

- The title is quite technical, making it difficult to guess what the article is about at first glance (for instance, it does not mention "soil heterogeneity"). I do not have any other suggestions, but the article would benefit from a clearer title.

We will think about another title, which clarifies the topic of our study better.

- I.101: The principles to which the authors are referring to are unclear. Maybe just keep the following sentence (I.102) or combine both?

We will rephrase the sentences as following: The vertical scheme of the model essentially couples soil thermal and hydrological processes through (i) phase change of soil water (Ekici et al., 2014) and (ii) changes in soil thermal properties due to soil moisture (Bonan, 2019).

- I.108-109: Could you give the soil layer thickness for all layers?

The layer thickness of DynSoM is user-adjustable. It has a fine resolution in the upper soil (as deep as the user choses) and then increasing layer thicknesses up to a depth of 50m. In our case (as we have chosen a 10cm resolution for the uppermost meter), our layer thicknesses are as following: 10cm 10cm,... We will add the layer thickness here.

- l.111-112: Bulk density is missing from the list (needed in pedotransfer functions of Wösten at al. (1999)).

DynSoM calculates bulk density based on the given information about soil texture, i.e. the fractions of clay and silt and organic matter content, based on the equations published by Martín et al. (2017) based on Leonavicicute (2000) The precise equations are included in the published code.

- I.115: What about the other soil parameters besides soil porosity?

In the pedotransfer functions, the following parameters are considered: van Genuchten parameters (alpha, m and n), residual porosity, saturated conductivity and from Cosby parameters (Cosby et al. 1984, saturated matric potential and the b parameter). In addition, the pedotransfer function of the code includes the calculation of the field capacity matric potential, fiels capacity effective saturation and field capacity moisture (all based in van Genuchten parameters), as well as initial porosity water and ice based on the calculated porosity and input fractions. We will add tables showing the range of the most relevant soil parameters to the SI.

- I.118: The variables driving the model should be specified. Also specify how rainfall and snowfall are separated from total precipitation.

We will add this to the SI.

- l.140 : Please provide the equations used to calculate soil thermal conductivity and capacity as these parameters largely control soil temperature dynamics.

We will add this to the SI.

- Given the crucial role of soil water content and frozen/unfrozen fractions on soil temperature, the soil hydrological scheme should be described (including boundary conditions).

We will add this to the SI

- l.156 "The heat is recalculated to a skin layer/surface temperature" is not very precise and should be clarified.

We will clarify this, but for short, we used the approach that is presented in Bonan (2019), chapter 7, section 7.3. Prominently equation 7.17 with sets the boundary condition for the soil temperature numerical solution (table 7.1).

- l.160-169: The equations governing snow cover dynamics should be provided, along with the values of snow conductivity, capacity and any other parameters controlling snow dynamics.

We will add this to the SI. See also Beer et al. (2018).

- l.174: For clarity, please recall that the soil column extends below 1m. If I have understood the code correctly, the soil column is 50m deep, but this should be clearly explained in the manuscript.

Yes, the total depth is 50m. We will clarify this.

- I.175: Which soil properties are used at depths below 1m?

We used the same properties as for the deep soil/C horizon. We will clarify this.

- Fig.1: The vertical axis needs a unit/label.

We will add them.

- l.178: Vegetation growth is excluded but it is unclear whether vegetation is present in the simulations. If so, please describe the type of vegetation present.

Vegetation is turned off and therefore absence. We simulated bare soil and also used bare soil properties (e.g. albedo). We will clarify this.

- Table 2: The percentages on the left do not add up to 100% for A-B.

Sorry, that's a typo. The A-B has 57.4% of soil. We will change this. We will also change the sand content of the OA in the right column to 3.25%.

- I.188: The authors should provide evidence that the soil temperatures and soil moisture are close to equilibrium after the 50-year spinup (with no long-term drift).

We have chosen a 50-years spin-up period, as we found this as a typical spin-up time for geophysical soil models that do not consider vegetation/a vegetated ecosystem, which would have needed to grow and find its equilibrium. In our preliminary analysis for the spin up length we have not observed sensible long-term remaining trends after the 20 first years spin up. Unfortunately, we haven't stored

the spin-up simulations as part of the provided dataset, as we considered the information of the complete spin-ups is not relevant enough to justify the storage that it requires. Consequently we are unable to prove this with our current simulations that are provided at zenodo. If the editor considers this information to be critical, we can create an new dataset containing the full spin-up result series.

- l.191-193: Analysing a single year does not produce robust quantitative results, as these depend on the atmospheric forcing of that particular year (e.g. the amount of precipitation influencing both snow cover and soil thermal properties via soil moisture). The aggregation error could be different if a different year had been chosen. Why not use a longer time period? I do not understand the need to "preserve the current atmospheric signal" (I.192). The results would be more robust if a longer period was included in the analysis, for example daily or monthly averages over 10 years, or the analysis of a longer timeseries if the authors prefer not to use time averages.

As described above, we did not want to describe the soil temperature at a specific site and validate the model, where a single year would be definitely not sufficient (besides the fact that we do not have measurement data), but to investigate the effect of soil heterogeneity and heterogeneity induced lateral heat fluxes on simulated soil temperatures (given the same atmospheric forcing). For our analysis, we used a single year to represent a "snap-shot" of the differences in soil simulations, which are solely caused by either soil heterogeneity or soil heterogeneity and heterogeneity induced lateral heat fluxes. The inclusion of several years, and especially the averaging over several years, would smooth and out-average the differences that we war aiming to explore, because the temporal differences between years will always outcompete soil spatial heterogeneity on such small distances. However, to show that the general pattern that we found by comparing simulations of a single year does not depend on the year itself, we will add figures for other exemplary years from our simultiont to the SI.

- l.201 : Please mention how θhet is calculated. Horizon-wise average?

The values in table 3 are spatially averaged for topsoil (0-20cm), subsoil (20-80cm) and deep soil (80-100cm), which means that they show the mean over all ten columns and within the given depths, and for the entire soil (0-100cm), again for all ten columns. See figure 1 for clarification.

- I.203 "In general, the horizon-wise averaged heat conductivity is lower in the heterogeneous soil than in the homogeneous soil"  $\rightarrow$  This statement is not supported by the data in Table 3 where  $\kappa$ het >  $\kappa$ hom for both the subsoil and the total soil.

This is true and misleading. We rephrase the sentence.

- l.208-210: I do not think it can be concluded from Fig.A1c&e that "the calculated albedo is hardly affected". The authors should provide data to support this statement.

We will add numbers here.

- I.234 : What does "per day" mean ? I suppose it refers to daily values but this should be clarified.

Yes. We will clarify this.

- 1.255: "at the simulated distance of 1m"  $\rightarrow$  unclear what this refers to.

We will rephrase this as following: ... , which exceeds  $4^{\circ}$ C on a daily basis and  $2^{\circ}$ C on a monthly basis across the simulated distance of 1m,...

- I.257 : "After freezing" → unclear, please precise.

We will clarify this as following: After freezing, i.e. when the soil is fully frozen up to 1m depth,...

- Using DOY in Fig.2 and months in Fig.3 makes it difficult to compare the two figures. Adding months to Fig.2 would improve understanding.

We agree that including months to these figures will make them better comparable to the following monthly analysis. However, we found DOY axes more convenient, when analyzing these timeseries themselves and time shifts by days. So, we will keep them, but try to add months.

- 1.276: "reflecting differences in soil moisture content that affect surface albedo"  $\rightarrow$  albedo data are missing to support this statement.

We will add some information (table or figure) to the SI.

- I.300: What does "from below" refer to? The freezing front?

We will rephrase this as following: During this week the soil freezes from both sides (fig. 3), i.e. from the surface, which is strongly cooled by the atmosphere, and from colder frozen layers below.

- Fig.4 : A statistical test, such as a t-test, is required in order to assess the significance of soil temperature differences.

If necessary, we can add the outcome of a t test, but the differences are small, and indeed not significant, which we also did not say here. We observed a clear pattern, linking OM content to larger differences in soil temperatures between model simulations that are modified by lateral fluxes. But neither the initial heterogeneity induced differences (M1-M0), nor the reduction (O horizon grid cells)/increase (OA, A, B, C horizon grid cells) of differences due to lateral fluxes (M2-M0 vs M2-M0) are significant.

- Fig.4: The difference in soil temperature is likely to be dominated by snow and ice in winter, resulting in values close to zero from February to June. Maybe focusing on the period when soil temperature differences are significant (July to January) would help to assess the effect of OM distribution on these differences, and would avoid pulling the median down artificially due to the near-zero ΔTsoil during half of the year?

The reviewer is right, that the observed full-year differences are resulting from close to zero differences within the first half of the year, but we think that the fact, that second-half year differences are still notable in full-year differences is noteworthy. We could artificially boost the effect by filtering the data, but we decided not to do this, because (1) OM content has also an effect on frozen soil properties, which we did not want to take out, and (2) the boosted effect would not be representative for the full year. However, we could aggregate the data seasonally (DJF, MAM, JJA, SON) and provide the figures in the SI.

- Section 3.4 lacks quantification. Statistical metrics and tests should be used.

As said before, the differences, and also not the change in differences due to lateral fluxes is not significant, and we never said it is. We observed a relationship between OM content and differences in simulated soil temperatures between a homogeneous and a heterogeneous soil, which is different for a model without and with lateral fluxes. But given our very small domain, the differences are small and not significant. We also show both, box whisker plots (incl. median, quartile ranges) of differences and mean differences, to be totally clear about the fact that (1) medians are close to zero and inner-quartile ranges largely overlap, which means that differences are not significant, and (2) mean values, which deviate stronger, are largely influenced by few outliers, which would also not show any significance, but need to be acknowledged.

- I.315-317 : "By aggregating...lower OM content"  $\rightarrow$  not sure if this is significant. A statistical test is needed.

We will rephrase this, since it was not meant to be a significance statement, but only compares grid cells (according to figure 1) by their OM content.

- I.317-318 : "this relationship...from M2 to M0" → based on which metrics?

We will rephrase this, because this is also only meant qualitatively, and not quantitatively, because we could not derive a clear (mathematical) relationship.

- I.356-357: This is counter-intuitive as OM generally insulates the ground against warmer air temperatures in permafrost regions (Zhu 2019, Loranty 2018). However this can be explained by the higher thermal conductivity in columns with a higher OM content, due to higher soil moisture, as the authors explain. A figure showing the 2D evolution of soil moisture (vertical and horizontal, similar to Fig.3) would support this statement.

Since DynSoM tends to be wet and (almost) saturated (see fig. A1), a figure showing soil moisture (water+ice) differences similar to figure 3 (or figure A2) indeed basically shows the difference in soil porosity, i.e. soil moisture is higher in the heterogenous soil, where the heterogeneous soil contains more OM, and lower, where the heterogeneous soil contains less OM than the homogeneous soil (see figure below this text). Differences are constant in frozen soil (Nov-June), and only differ slightly, where soil is thawed in summer. The differences in the active layer thickness are visible by the small negative deviation in around 0.5m depth in summer, where the model with homogeneous soil is frozen, but the models with heterogeneous soil thaw 10-20cm deeper. We can add this figure to the SI, but we do not see a major benefit here.

- I.375-376 : "The only time when snow actually has an effect on simulated surface heat fluxes is at the end of the snow melt period" → rather on the difference of surface heat fluxes between homogeneous and heterogeneous configurations than on surface heat fluxes themselves (which are always impacted by snow) ?

This is true. We will rephrase this.

- I.409 : "which will be consistent for any aggregation error."  $\rightarrow$  I do not think this conclusion can be drawn from a single-site study. Such a conclusion would require simulations for other sites, as the

amplification of the aggregation error by lateral fluxes probably depends on the spatial distribution of soil texture and OM content.

We will rephrase this as following: "which will likely be consistent for any aggregation error."

- I.443-446: I do not understand why the amplification of the aggregation error by lateral fluxes provides more confidence that these results would remain for other sites.

We will better justify our conclusion.

- I.448 : In the context of land surface modelling, there are no global maps of soil texture and OM at a 10 cm resolution. Could you briefly describe how these results could help improving land surface models?

We will add a section to the discussion about this. Just shortly, our results clearly show that soil heterogeneity has an impact on simulation results, and that they are not out-balanced by lateral fluxes, but rather amplified. With decreasing model resolution, heterogeneity becomes more obvious, and heterogeneity induced lateral fluxes will become more important. Our example is of course a rather extreme one, since most LSMs run on scales of (hundreds of) kilometers, but we want to emphasize that as soon as model developers think about decreasing model resolution, they have also to think about lateral fluxes and acknowledge their influence on simulation results.

- I.453 : "we assume that it will remain at other sites and under changed environmental conditions" → I do not think this assumption can be made based on a single-site study.

We will rephrase this as following: "we assume that it will likely remain at other sites and under changed environmental conditions"

- Fig. A3 and Fig.A4 are not referenced in the main text. They should either be included or removed.

We will add references.

**Technical corrections**

- Bonan 2019 is cited for various aspects of the model and it would be helpful to provide more specific references for each citation (e.g. refer to the relevant chapter in the book).

We will add chapter numbers and/or refer to equations

- l.105 : Please give the section number.

We will add chapter numbers and/or refer to equations

- I.123 : "phase change" → water phase change

We will change this

- I.126: Please refer to the specific chapter/equations from Bonan (2019).

We will add chapter numbers and/or refer to equations

- I.131-134: Please precise the units.

We will add this.

- l.136 : Please give the section number.

We will add chapter numbers and/or refer to equations

- I.138: Please precise the units.

We will add this.

- I.154: Please refer to the specific chapter/equations from Bonan (2019).

We will add chapter numbers and/or refer to equations

- l.171: Please specify which figure from Gentsch et al. (2015) you are referring to.

We will add this information.

- Table 2: Please define O, A, B and C in the table legend.

We will add this.

- Figure 1 : Please precise in the legend what the red dash-dot lines refer to (separation topsoil, subsoil and deep soil?).

We will add this.

- Table 3: Please add units.

We will add this.

- Fig.2 (b): The peak of maximum difference looks cropped.

It is. We decided to do so, to focus on the smaller mean differences, instead of the maximal range, which happens only in a short period of time. We will add the exact numbers to the text.

- I.234: "In general" should be removed.

We will remove this.

- deVrese et al. Paper is now published and is no longer a preprint.

We will change this.

**References**

Loranty, M. M., Abbott, B. W., Blok, D., Douglas, T. A., Epstein, H. E., Forbes, B. C., Jones, B. M., Kholodov, A. L., Kropp, H., Malhotra, A., Mamet, S. D., Myers-Smith, I. H., Natali, S. M., O'Donnell, J. A., Phoenix, G. K., Rocha, A. V., Sonnentag, O., Tape, K. D., and Walker, D. A.: Reviews and syntheses: Changing ecosystem influences on soil thermal regimes in northern high-latitude permafrost regions, Biogeosciences, 15, 5287–5313, https://doi.org/10.5194/bg-15-5287-2018, 2018.

Wösten, J., Lilly, A., Nemes, A., and Le Bas, C.: Development and use of a database of hydraulic properties of European soils, Geoderma, 90, 169–185, 1999.

Zhu, D., Ciais, P., Krinner, G. et al. Controls of soil organic matter on soil thermal dynamics in the northern high latitudes. Nat Commun 10, 3172 (2019). https://doi.org/10.1038/s41467-019-11103-1

Beer, C., Porada, P., Ekici, A., & Brakebusch, M. (2018). Effects of short-term variability of meteorological variables on soil temperature in permafrost regions. The Cryosphere, 12(2), 741-757. https://doi.org/10.5194/tc-12-741-2018

Leonaviciute, N. (2000). Predicting soil bulk and particle densities by pedotransfer functions from existing soil data in Lithuania. Geografijos metraštis, 33, 7-330.

Martín, M. Á., Reyes, M., & Taguas, F. J. (2017). Estimating soil bulk density with information metrics of soil texture. Geoderma, 287, 66-70. https://doi.org/10.1016/j.geoderma.2016.09.008

---

## Author Comment (AC4)

The manuscript presents an attempt to quantify the significance of the treatment of small-scale variations in soil properties on the simulated soil and surface temperatures, as well as the surface heat fluxes. The effects of heterogeneity and their impact on larger-scale averages are undoubtedly intriguing topics that warrant the attention of the modeling community. However, I think the manuscript would benefit from some revisions of the description of the model and analysis, and expanding of analysis to include the effects of interannual variability.

We are grateful to the reviewer for emphasizing the importance of this study and the constructive feedback. We will address the mentioned problems point by point.

To start, I would suggest changing the title of the manuscript. First, for readers not intimately familiar with the permafrost features and process, it is not at all obvious what "non-sorted circles" in the title refers to. As a very minimum, the title should make it clear that the study is specific to the permafrost processes. Second, the focus of the study is on the differences in results of several modeling approaches, but to the casual reader the term "aggregation error" implies comparison with observations (or the perfect representation of the processes), which is not the focus of the manuscript. I recommend avoiding this term, at least in the title.

We will think about another title, which will make the topic of our study clearer.

I think another problem is that the description of the model misses an essential part: the method used to calculate surface turbulent fluxes. Clearly, on such a small horizontal scale of ~10 cm that the described model uses, the Monin-Obukhov Similarity Theory (MOST) approach commonly employed in large-scale mosaic schemes would not work because a number of assumptions important for the MOST applicability are violated. Therefore, it is essential to describe what alternative approach was used to calculate surface fluxes, especially given that a significant portion of the manuscript is devoted to the analysis of differences in turbulent fluxes and energy balance. Without that, it is very hard to judge the validity of the results.

The method used to calculate the surface turbulent flues is indeed Monin-Obukhov Similarity Theory, but from our perspective, there are three reasons, which we can use MOST for our model: (1) All MOST assumptions are limited to the "atmospheric" part of our model that is only applied vertically (1D). Derived energy fluxes are then used for all columns individually. (2) For this specific study, we used only "bare soil" and neglected any topographical differences, which means no differences in height and/or roughness between columns. Snow is added to the soil scheme, which does not affect surface height in this model configuration. Consequently, soil roughness versus the reference height of 2m (lowest level taken from CRUNCEP data) is well enough the recommended value of 50. However, (3) if roughness differences) due to vegetation or topographical differences) are present, DynSoM couples MOST with a roughness sublayer parameterization following Harman and Finnigan (2007, 2008). We will clarify this in the model description.

The description also seems to contradict itself, saying in section 2.1.1 that "a prescribed skin temperature (see following section) serves as the upper boundary condition", while equation (3) implies that the surface skin temperature is calculated given the atmospheric meteorological forcing and prognostic equations of water and energy balance in the soil. I think providing more details about the calculation of surface fluxes and energy balance would help to resolve this confusion.

We took this method from Bonan (2019) and will refer to the specific chapter here.

In the section describing the results of the simulations, the authors chose to limit the analysis of the results to just one year, but the motivation is not entirely clear to me: the text in lines 191-192 says "to preserve the current atmospheric forcing signal, which is strongly superimposed when another

averaging method is used." I am not sure about the meaning of this phrase; especially since it somewhat contradicts the one-week smoothing applied to the results to "avoid overly fuzzy near-surface" values. I think analyzing and presenting the statistics across all available years of forcing is essential to take into account the interannual variability and to have confidence in the robustness of the results.

This does not, of course, preclude using the results from one year (or one season) as an example illustrating physical processes, if necessary.

We agree that founding our study one a single year limits its significance. We will the full timeseries of figure 2 to the SI (see upper figure below this text) to show that the general observed pattern, i.e. simulation differences that are solely caused by soil heterogeneity and their increase by lateral heat fluxes, is not dependent on single years, whereby of course the absolute differences between models (inter-model ranges) differ across years. For figure clarity, we will leave out the intra-model ranges here, whereby these obviously also differ across years. To keep this information, we will replace figure 4 (see lower figure below this text as example; as well as related figures in the SI) by figures showing boxplots for the entire period and rewrite section 3.4 accordingly.

In the description of the energy-related fluxes and balances, the authors frequently use the confusing phrase "kW/m^2 per day": the fluxes are typically measured in W/m^2, and it is absolutely not clear what "per day" refers to. Similarly confusing is the phrase "°C per day" in the description of the range of temperatures.

We will change this.

Assuming that my understanding of the units used in the analysis is correct, some of the energy balance numbers seem to be unreasonably large. For example, on lines 221-223, the manuscript says "the total simulated annual heat budget ... (M0: 13.7kW/m2, M1: 14.1kW/m2, M2: 13.5kW/m2)". Of course, the total long-term average energy balance at the surface of the well-spun-up land model should be close to zero, so it is not clear what these numbers represent.

I can only assume that these results represent the annual average sum of sensible and latent heat fluxes (which should compensate radiative fluxes), but even then the numbers seem excessively high. For comparison, the solar radiation incident on the area perpendicular to the sun rays at the top of the Earth atmosphere is ~1360 W/m2. It is not clear how it is possible that the annual heat balance at the site (~69N latitude) can be so large, given attenuating factors due to site latitude, annual averaging, and absorption/reflection/scattering by the atmosphere and clouds. Unless this is a typo, an explanation must be provided. Likewise, the intra-model differences in heat fluxes are on the order of hundreds of W/m2: that of course is not impossible on the short time scale, but would strongly depend on the way the turbulent fluxes are calculated, and needs to be discussed.

The reviewer is right. This is a unit error. All energy fluxes are actually given in  $W/m^2$ . We will change this in the manuscript.

In figure A1, soil water content is measured in kg/kg; this is kilogram of water per kilogram of what dry soil or wet soil? Or per dry soil+ice? Why the commonly accepted definition of volumetric water content is not used?

It is soil water (A1c & d) and soil ice (A1e & f) in kg per kg dry soil accordingly to Bonan (2019), chapter 16. We will clarify this.

**Technical comments:**

Line 38, and elsewhere replace "snow height" with "snow depth"

We will changes this.

L 171: Provide coordinates for Cherskii site

We will add this

L 178: Typo: "growthto" should be "growth to"

We will change this.

L 203: Replace "horizon-wise averaged" with "horizontally averaged"

We disagree here, because "horizontally averaged" would only imply the horizontal averaging over single rows, which have (in our model configuration) a vertical extension of 10cm, whereas the "horizon-wise averaging" that we applied implies a larger vertical extension (in our model configuration).

Caption of figure 2, and elsewhere: does T\_surf refer to the temperature of the soil surface, or the surface interacting with the atmosphere (i.e. surface of the snowpack if present and soil surface T otherwise)?

Because snow is part of the soil in DynSoM T\_surf refers to the surface that is directly interacting with the atmosphere. We will clarify this.

Caption of figure 2: "T\_soil, all depths" — does it mean averaged over entire soil column?

Averaged until the depth of 1m. We will clarify this.